# Ratiometric measurement of MAM Ca²⁺ dynamics using a modified CalfluxVTN

Eunbyul Cho[1,2], Youngsik Woo ®[1,2] ✉, Yeongjun Suh ®[1], Bo Kyoung Suh[1], Soo Jeong Kim ®[1], Truong Thi My Nhung[1], Jin Yeong Yoo[1], Tran Diem Nghi[1], Su Been Lee ®[1], Dong Jin Mun[1] & Sang Ki Park ®[1] ✉

Mitochondria-associated ER membrane (MAM) is a structure where these calcium-regulating organelles form close physical contact sites for efficient Ca²⁺ crosstalk. Despite the central importance of MAM Ca²⁺ dynamics in diverse biological processes, directly and specifically measuring Ca²⁺ concentrations inside MAM is technically challenging. Here, we develop MAM-Calflux, a MAM-specific BRET-based Ca²⁺ indicator. The successful application of the bimolecular fluorescence complementation (BiFC) concept highlights Ca²⁺-responsive BRET signals in MAM. The BiFC strategy imparts dual functionality as a Ca²⁺ indicator and quantitative structural marker specific for MAM. As a ratiometric Ca²⁺ indicator, MAM-Calflux estimates steady-state MAM Ca²⁺ levels. Finally, it enables the visualization of uneven intracellular distribution of MAM Ca²⁺ and the elucidation of abnormally accumulated MAM Ca²⁺ from the neurons of Parkinson's disease mouse model in both steady-state and stimulated conditions. Therefore, we propose that MAM-Calflux can be a versatile tool for ratiometrically measuring dynamic inter-organellar Ca²⁺ communication.

Mitochondria and endoplasmic reticulum (ER) form close contact sites spanning ~10–30 nm and are called mitochondria-associated ER membranes (MAM) or mitochondria-ER contacts (MERC)[1,2]. MAM is a signaling hub that governs ER-mitochondria Ca²⁺ transfer[3–6], lipid synthesis and exchange[7], and mitochondrial structure dynamics[2] to regulate autophagy[4,8], ER stress response[9], inflammasome activation[10], and apoptosis[11]. Accumulating evidence implies that MAM is rearranged in response to diseases such as Parkinson's disease (PD)[12,13], Alzheimer's disease (AD)[14,15], amyotrophic lateral sclerosis (ALS)[16,17], and type 2 diabetes mellitus (T2DM)[18]; yet the underlying mechanisms are largely unexplored.

MAM serves as a ground for dynamic Ca²⁺ transfer from the ER to mitochondria, thus playing an important role in diverse biological processes. Ca²⁺ indicators, which are located at the mitochondrial matrix or cytosolic part of the outer mitochondrial membrane (OMM), determined that, upon Ca²⁺ release from the ER, the mitochondria are then exposed to a higher concentration of Ca²⁺ compared with that in the cytosol; this microdomain with high-Ca²⁺ concentration is the MAM[3,14]. The MAM is concentrated with several proteins responsible for MAM Ca²⁺ transfer, including inositol triphosphate receptors (IP3Rs) and polycystin-2 (PKD2), which are involved in Ca²⁺ release from the ER, and voltage-dependent anion channels (VDACs), which are involved in Ca²⁺ influx into the mitochondria. In addition, the MAM structure is maintained by tethering proteins such as glucose-regulated protein 75 (GRP75) and DJ-1[1,13], which also participate in regulating MAM Ca²⁺ transfer. Accumulated Ca²⁺ on the mitochondrial surface activates enzymes responsible for cellular bioenergetics and mitochondrial transport[9,19,20]. MAM Ca²⁺ transfer is also important for inhibiting autophagy[4,8], stimulating Drp1-dependent mitochondrial division and pro-apoptotic signaling[2,21], and buffering local cytosolic Ca²⁺ concentration[5]. Despite its importance, there is a limitation in the methodology used to directly and specifically measure MAM Ca²⁺, resulting in a poor understanding of MAM Ca²⁺ physiology.

[1]Department of Life Sciences, Pohang University of Science and Technology, Pohang 37673, Republic of Korea. [2]These authors contributed equally: Eunbyul Cho, Youngsik Woo. ✉e-mail: youngsik.woo@postech.ac.kr; skpark@postech.ac.kr

Genetically encoded calcium indicators (GECIs) are powerful tools for understanding $Ca^{2+}$-mediated cellular functions, particularly in cell biology and neuroscience. GCaMP, one of the most abundantly utilized GECI, is composed of a calmodulin-binding sequence of myosin light chain kinase (M13), circularly permutated green fluorescent protein (GFP), and calmodulin. The binding of $Ca^{2+}$ to calmodulin and M13 induces conformational changes, which increase the GFP fluorescence intensity[22]. Over the past decades, significant progress has been made in the development of GECIs with longer wavelengths, higher signal-to-noise ratios, and faster kinetics[23,24]. In addition, ratiometric versions of GECIs were invented to overcome the limitations of GCaMPs whose basal signal intensities are multiplied in an expression-level-dependent manner. Ratiometric GECIs enable the quantitation of steady-state $Ca^{2+}$ concentrations, such as in GCaMP-R based on the GCaMP/RFP ratio[25], Twitch based on fluorescence resonance energy transfer (FRET)[26], and Nanolantern or CalfluxVTN based on bioluminescence resonance energy transfer (BRET)[27]. Efforts have been made to understand intracellular $Ca^{2+}$ distribution by developing organelle- or compartment-specific GECIs, such as for mitochondria[28], ER[29], and synapses[30]. However, limitations exist in directly and specifically investigating MAM $Ca^{2+}$, and hence, indirect estimations are made by comparing $Ca^{2+}$ dynamics separately measured from the ER and mitochondria[6,31]. Thus, the development of a MAM-specific $Ca^{2+}$ sensor represents the effort to better understand the physiology of MAM $Ca^{2+}$ dynamics.

In this study, we devise a modified version of CalfluxVTN which is specific for MAM, named MAM-Calflux. MAM-Calflux can directly probe MAM-specific $Ca^{2+}$ concentration by applying bimolecular fluorescence complementation (BiFC). Venus fluorescence is only operative where the ER and mitochondria are closely tethered, allowing the MAM-specific observation of $Ca^{2+}$-responsive BRET signals. The specificity of the Venus to the MAM also enables the use of this sensor as a quantitative structural marker, facilitating the dual functionality. MAM-Calflux demonstrates an uneven distribution of MAM $Ca^{2+}$ dynamics. Furthermore, we apply this sensor to estimate MAM $Ca^{2+}$ dynamics in the neurons of multiple neurodegenerative disease mouse models.

## Results

### MAM-Calflux, a BRET-based MAM-specific calcium indicator

To establish a MAM-specific $Ca^{2+}$ indicator, we applied the concept of bimolecular fluorescence complementation (BiFC)[32] to a BRET-based $Ca^{2+}$ indicator, CalfluxVTN[27], with organelle targeting sequences. CalfluxVTN was separated into two fragments: 1–173 aa Venus fluorescence domain (VN173) and 156–228 aa Venus domain (VC155) conjugated with troponin C and Nanoluciferase (NanoLuc) domains. Linker sequences of optimized lengths were used to insert the ER-targeting sequence (Sac1 521–587 aa) and mitochondria-targeting sequence (AKAP 1–30 aa) into VN173 and VC155 fragments, respectively (Fig. 1a). According to the principle of BiFC assays, the Venus fragment is only fused at the sites where the mitochondrial and ER membranes are closely juxtaposed. This is to ensure that the BRET signal of MAM-Calflux is visible only at the MAM after binding of $Ca^{2+}$ to troponin C, which triggers a conformational change due to the close apposition between NanoLuc and fused Venus. Thus, MAM-Calflux functions as a MAM-specific $Ca^{2+}$ sensor (Fig. 1b). Additionally, after Venus is fused at the MAM, it can be visualized by the excitation-based fluorescence imaging instead of the bioluminescence- and $Ca^{2+}$-dependent BRET. Therefore, MAM-Calflux can also be utilized as a MAM-specific structural marker using Venus fluorescence imaging without a NanoLuc substrate, thus displaying dual functionality (Fig. 1c).

First, we confirmed the organelle-specific localization of each of the two MAM-Calflux fragments. The expression of ER-targeted VN173

(MAM-Calflux-N) with the soluble form of VC155 resulted in a Venus fluorescent pattern that significantly overlapped with that of Sec61β-mCherry, an ER marker (Supplementary Fig. 1a, b). When mitochondria-targeted VC155 with troponin C and NanoLuc (MAM-Calflux-C) was transfected with the soluble form of VN173, Venus distribution significantly overlapped with that of TOM20, a mitochondrial marker (Supplementary Fig. 1c, d). Finally, co-expression of MAM-Calflux-N and MAM-Calflux-C resulted in the localization of Venus at the site where ER and mitochondrial markers overlapped, confirming the MAM-specific localization of MAM-Calflux (Fig. 1d–f). Since the self-assembly between fragments in the BiFC strategy was reported[33], which can affect MAM formation, we optimized the experimental condition for the MAM-Calflux. MAM-Calflux expression did not affect MAM structures by 17 h after transfection (Supplementary Fig. 1e), but enhanced MAM structure by 48 h, providing optimal experimental time window (within 16–24 h after transfection) for measurement of MAM $Ca^{2+}$ using MAM-Calflux.

To validate the $Ca^{2+}$ response of MAM-Calflux, we measured the intensities of luminescence and BRET-based Venus fluorescence with or without $Ca^{2+}$ using the spectrometry after treatment with furimazine, a substrate for NanoLuc, and ionomycin, a membrane-permeabilizing agent. In HeLa cells, either the co-expression of MAM-Calflux-N and MAM-Calflux-C or the single expression of MAM-Calflux-C resulted in a bioluminescent signal peak at 450–470 nm wavelength, confirming that NanoLuc functionality was maintained despite Venus cleavage (Supplementary Fig. 2a). MAM-Calflux had a significant emission peak at 520–530 nm wavelength, which was diminished by the $Ca^{2+}$ chelator EGTA, indicating the existence of a $Ca^{2+}$-responsive BRET signal (Supplementary Fig. 2a, b). MAM-Calflux-C, which contains intact NanoLuc but lacks the functional Venus, only had a bioluminescent signal without the BRET signal, implying that the BRET signal necessitates the fusion of Venus fragments. Rising BRET signals were observed as $CaCl_2$ concentrations increased, confirming the capacity of MAM-Calflux to detect MAM $Ca^{2+}$ (Supplementary Fig. 2c, d). The $Ca^{2+}$ response kinetics of the sensor was also determined in vitro using purified proteins with various concentrations of free-$Ca^{2+}$ via spectrometry (Fig. 1g, h).

Furthermore, we performed a microscopic image analysis of the BRET signal. Since the luminescence signal is much weaker than the fluorescence signal, we utilized an EMCCD camera with a beam splitter (dual-view) to acquire images of the bioluminescence (Em460) and BRET-based Venus (Em525) signals simultaneously in a single frame (Supplementary Fig. 3a–b). Since MAM-Calflux measures $Ca^{2+}$ levels in MAM regions, we quantitated the BRET ratio after masking MAM regions based on Venus excitation signals (Supplementary Fig. 3c). Time-lapse imaging verified that MAM-Calflux stably emitted Em460 and Em525 signals after ~2 min of furimazine treatment, which was maintained for at least 10 min (Supplementary Fig. 3d). Notably, even though the Em525 signal was significantly detected, it was negligible compared with the excitation-based Venus fluorescent signal (Venus excitation) (Supplementary Fig. 4). This allowed us to simultaneously observe the MAM structure based on Venus excitation and the MAM $Ca^{2+}$ concentration based on Em460/Em525 signals while the furimazine exists, emphasizing its dual functionality as a MAM marker and MAM $Ca^{2+}$ indicator.

### MAM-Calflux visualizes the changes in MAM-calcium dynamics

Next, we attempted to validate the MAM-Calflux response to changes in MAM $Ca^{2+}$. We treated the cells with histamine, which indirectly increases ER $Ca^{2+}$ release through IP3Rs[3], followed by $CaCl_2$ and ionomycin, which directly induce excessive calcium influx. Time-lapse imaging detected an immediate increase in the BRET ratio (Fig. 2a–c). Meanwhile, Venus excitation was not affected by histamine and $CaCl_2$/ionomycin, implying that the MAM structure was not

 

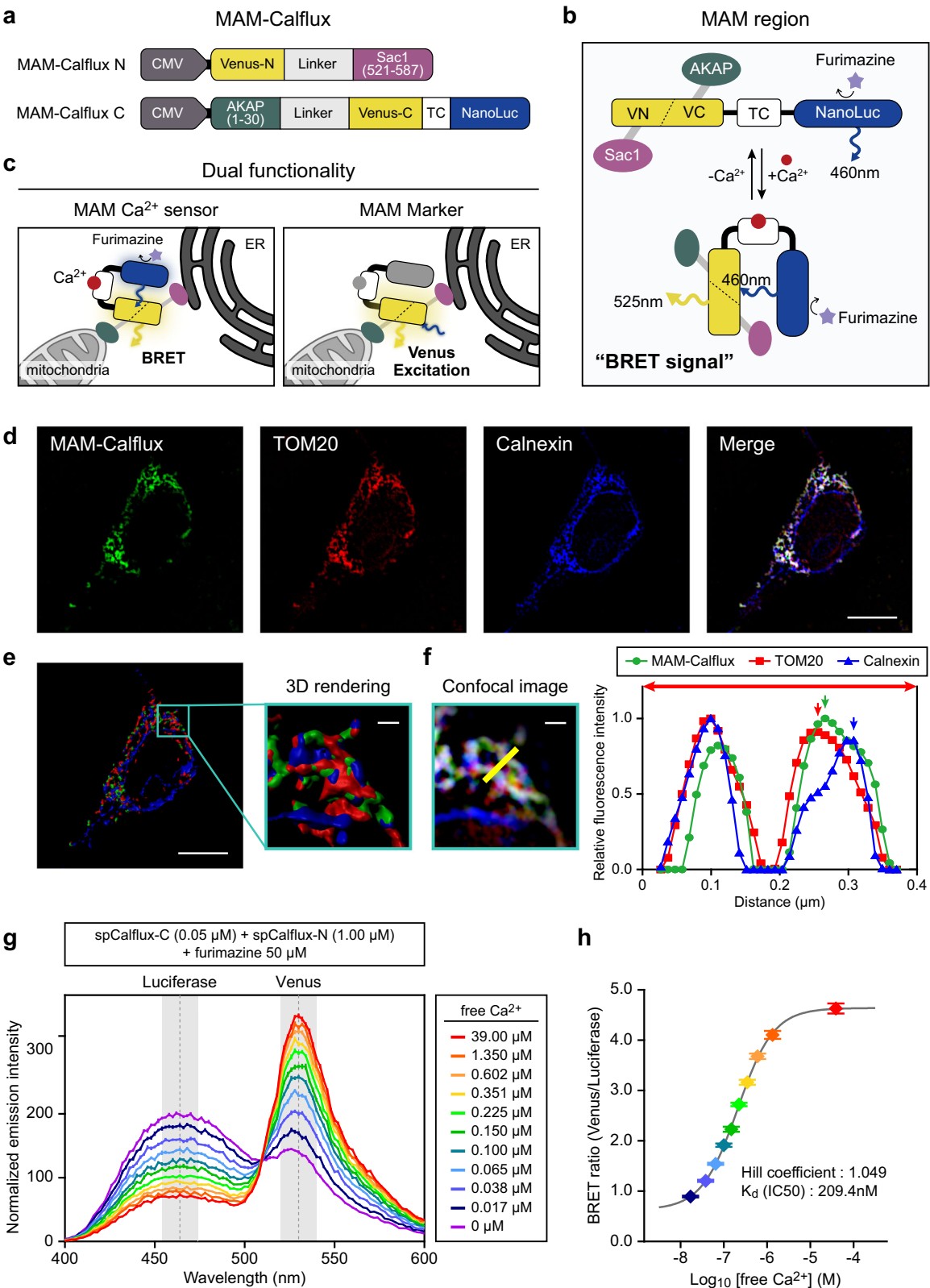

significantly affected and the increased BRET ratio was specific for $Ca^{2+}$ influx (Fig. 2d, e). The BRET ratio was restored within a few minutes after histamine treatment, while the increased BRET ratio after $CaCl_2$/ionomycin treatment was maintained for >3 min (until the end of imaging). This indicates that the restored BRET ratio after histamine treatment presumably demonstrated active $Ca^{2+}$ restoration in MAM.

## MAM-Calflux measures changes in the steady-state $Ca^{2+}$ level at MAM

Compared with single fluorescent protein-based calcium indicators, including GCaMPs, FRET-based or BRET-based calcium indicators have the advantage that they are relatively independent from their expression levels because $Ca^{2+}$ concentration is calculated as ratiometric signals[26]. This advantage allowed us to estimate steady-state MAM $Ca^{2+}$

**Fig. 1 | MAM-Calflux, a BRET-based MAM-specific calcium indicator. a** Schematic design of MAM-Calflux constructs. Based on the CalfluxVTN construct, Venus was divided into two fragments, Venus-N (1–173 aa of Venus, VN) and Venus-C (156–228 aa of Venus, VC). Venus-N was conjugated with ER-targeting sequence (521–587 aa of Sac1) and Venus-C with troponin C (TC), and Nanoluciferase (NanoLuc) was conjugated with mitochondria-targeting sequence (1–30 aa of AKAP). **b** Schematic representation of the working principle of MAM-Calflux function depending on $Ca^{2+}$ binding. At the MAM site, Venus-N and Venus-C associate with each other to form a functional fluorescence domain. NanoLuc emits a bioluminescence signal (460 nm) in the presence of the substrate (furimazine). When $Ca^{2+}$ binds to TC, the conformational change results in drawing the fused Venus and NanoLuc into close proximity, followed by energy transfer and emission of the Venus signal (525 nm). **c** A dual functionality of MAM-Calflux. In addition to its application in $Ca^{2+}$ sensing (left), the presence of functional Venus in the MAM region enables MAM-Calflux as

a MAM structural marker through direct laser excitation of Venus (right). **d–f** Intracellular localization of MAM-Calflux. Transfected HeLa cells were immunostained with TOM20 and Calnexin as markers for mitochondria and ER, respectively (**d**). The 3D rendering image (**e**) and the line profile plot (**f**) represent MAM-Calflux localization between mitochondria and ER. Data representative of three experimental repeats. **g–h** $Ca^{2+}$-dependent BRET from in vitro purified split-version of CalfluxVTN was measured using a microplate reader system. Luminescence wavelength scan (**g**) and the sigmoidal curve fitting of BRET ratios upon various free-$Ca^{2+}$ concentrations (**h**). ($n = 16, 16, 15, 16, 16, 12, 13, 16, 15, 16$ microplate wells for 39.00, 1.350, 0.602, 0.351, 0.225, 0.150, 0.100, 0.065, 0.038, 0.017 μM free-$Ca^{2+}$ groups, respectively, and $n = 16$ microplate wells for no $Ca^{2+}$ group). The scale bars represent 10 μm for **d** and (**e**, left) and 1 μm for (**e**, right) and (**f**). All results for **g**, **h** are presented as mean ± SEM. Source data from **f**–**h** are provided as a Source Data file.

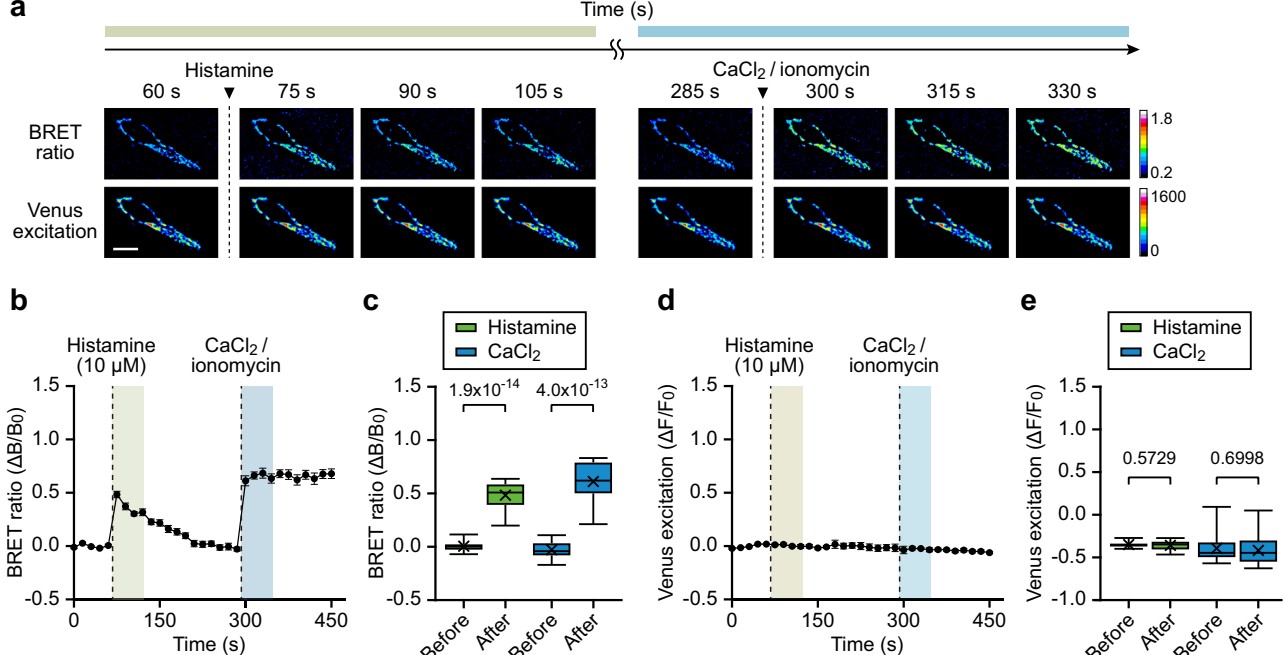

**Fig. 2 | MAM-Calflux visualizes the changes in MAM-specific calcium dynamics. a–e** Time-lapse imaging of MAM-Calflux with serial treatments of 10 μM histamine and 10 μM ionomycin with 1 mM $CaCl_2$. Representative images (**a**) and time-dependent plots for BRET ratio (**b**) and Venus excitation (**d**). Bar graphs represent the BRET ratio (**c**) and Venus excitation intensities (**e**) before and after histamine and $CaCl_2$/ionomycin treatments. ($n = 15$ cells for serial treatment group). The scale bar represents 20 μm. All results are presented as means ± SEM for **b**, **d** and box plots representing median and interquartile range with whiskers min/max value and the cross representing mean value for **c**, **e**. All *P* values were calculated using two-tailed Student's *t* test comparing before and after each treatment for **c**, **e**. Source data from **b**–**e** are provided as a Source Data file.

levels via MAM-Calflux imaging without any external $Ca^{2+}$ stimulation. Thus, we introduced the conditions that were reported to regulate MAM contact size and examined the basal levels of the BRET ratio and Venus excitation of the MAM $Ca^{2+}$ sensor and MAM marker, respectively. VAPB and PTPIP51 form a tethering complex that mediates the formation of the MAM structures[8,16]. The over-expression of VAPB and PTPIP51 increased both Venus excitation and BRET ratio in HeLa cells and primary cultured mouse neurons (Fig. 3a–d and Supplementary Fig. 5a–d), implying that both the MAM structure extent and MAM $Ca^{2+}$ level, respectively, were increased by this tethering complex. In addition, VAPB-PTPIP51 expression increased histamine-induced $Ca^{2+}$ influx in MAM and mitochondria without affecting cytosolic $Ca^{2+}$ influx and ER $Ca^{2+}$ release (Supplementary Fig. 5e–l and Supplementary Fig. 6). GRP75 is a chaperone tethering IP3R and VDAC1 mediating MAM $Ca^{2+}$ crosstalk[1]. The BRET signals and total Venus intensities were upregulated by the over-expression of GRP75 (Fig. 3e–h), meaning that GRP75 increased MAM $Ca^{2+}$ levels and MAM structure closeness but did not

change the MAM structure area. This was recapitulated by staurosporine (STS) 2 h treatment (Supplementary Fig. 7), which induces early phase of apoptosis along with increased intracellular $Ca^{2+}$ influx and MAM formation[34,35]. In contrast, the knockdown of a tethering protein MFN2 diminished basal MAM $Ca^{2+}$ levels with low amounts of MAM structures (Fig. 3i–l). These results support the capacity of MAM-Calflux to determine changes in steady-state MAM $Ca^{2+}$ level as well as in MAM structure.

## MAM-Calflux can determine the local distribution of MAM $Ca^{2+}$ dynamics

Targeting GECIs to the mitochondrial surface helped visualize the uneven $Ca^{2+}$ distribution on high-$Ca^{2+}$ microdomains, which are largely relevant to the MAM[14,36]. In addition, local $Ca^{2+}$ buffering at specific regions, especially the cell periphery and synapses, is regarded as an important role of mitochondria, ER, and MAM in maintaining their respective cellular functions[37–39]. However, whether MAM $Ca^{2+}$

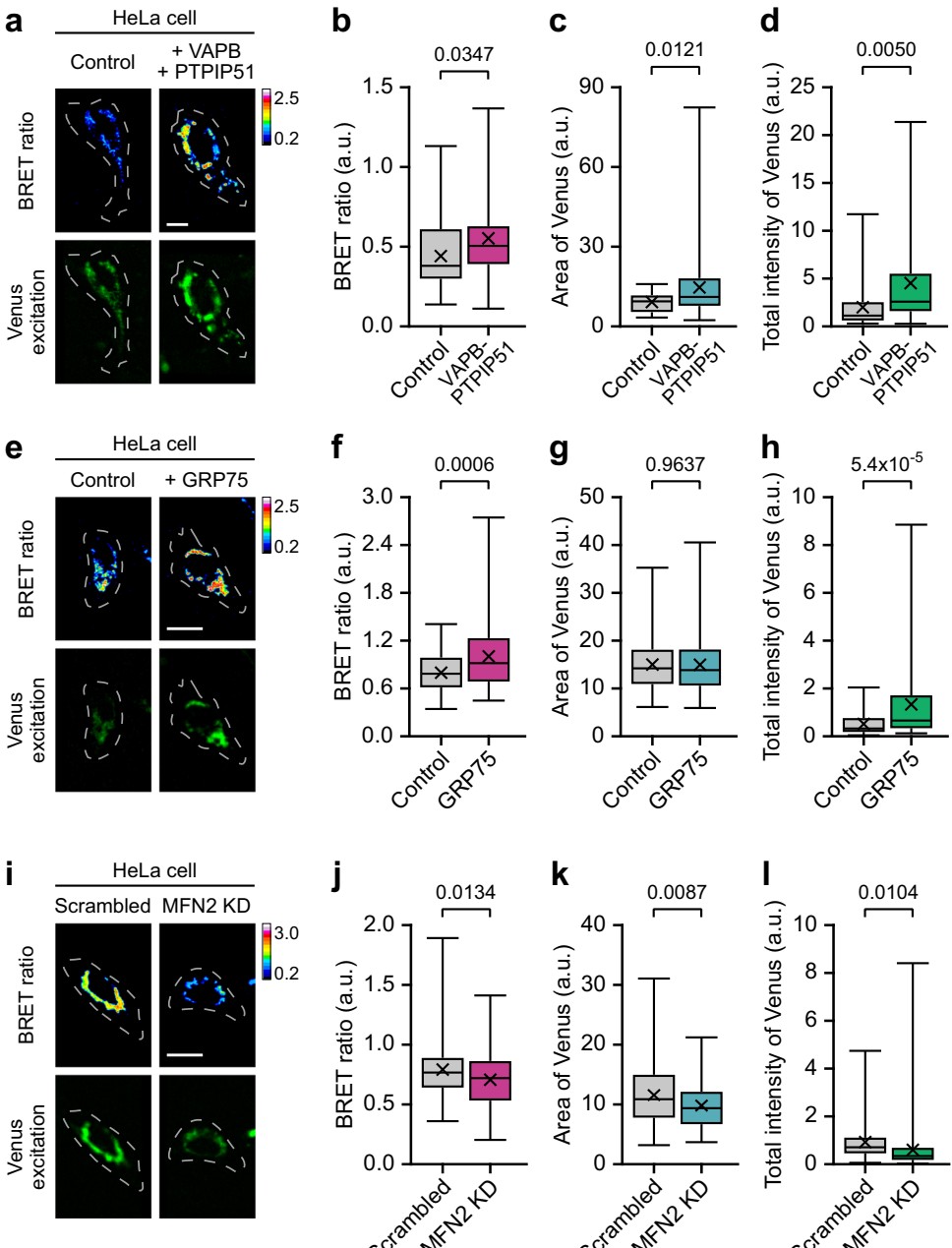

**Fig. 3 | MAM-Calflux measures changes in the steady-state Ca²⁺ level at MAM.**
**a–d** Increased MAM Ca²⁺ levels and enhanced MAM structure formation measured by MAM-Calflux upon over-expression of VAPB-PTPIP51 tethering complex in HeLa cells. Representative images (**a**) and bar graphs representing BRET ratio (**b**), MAM area (**c**), and total Venus excitation intensities (**d**). Dashed lines in **a** represent the cell morphologies. ($n = 45$ for control HeLa cells, $n = 33$ for VAPB-PTPIP51 expressing HeLa cells). **e–h** Increased MAM Ca²⁺ levels and intact MAM structure measured by MAM-Calflux upon over-expression of GRP75 in HeLa cells. Representative images (**e**) and bar graphs representing BRET ratio (**f**), MAM area (**g**), and total Venus excitation intensities (**h**). ($n = 68$ for control HeLa cells, $n = 82$ for GRP75

expressing HeLa cells). **i–l** Decreased MAM Ca²⁺ levels and reduced MAM structure formation measured by MAM-Calflux upon knockdown of MFN2 in HeLa cells. Representative images (**i**) and bar graphs representing BRET ratio (**j**), MAM area (**k**), and total Venus excitation intensities (**l**). ($n = 108$ for control HeLa cells, $n = 99$ for MFN2 knockdown HeLa cells). Dashed lines in **a**, **e**, and **i** outline the cell morphologies. The scale bars represent 10 μm. All results are presented as box plots representing the median and interquartile range with whiskers min/max value and the cross representing the mean value. All P-values were calculated using two-tailed Student's *t* test for **b–d**, **f–h**, and **j–l**. Source data from **b–d**, **f–h**, and **j–l** are provided as a Source Data file.

dynamics is locally determined independent of MAM structure has not yet been directly explored.

Based on the dual functionality of MAM-Calflux as a structural marker and a Ca²⁺ sensor, we hypothesized that MAM-Calflux can distinguish the intracellular localization of MAM structures and the distribution of Ca²⁺ concentrations among them. Line profile analysis in primary cultured neurons, which are highly polarized and necessitate the local distribution of organelles and Ca²⁺ concentration[40,41], identified

synchronized peak distributions of MAM Ca²⁺ intensity compared with that of MAM marker intensity (Supplementary Fig. 8a, b). Meanwhile, comparing MAM Ca²⁺ peak intensities between regions with corresponding MAM structural closeness, we found significant differences among them implying the local distribution of MAM Ca²⁺ dynamics among the MAM regions (Supplementary Fig. 8b). Furthermore, during time-lapse imaging of MAM-Calflux-expressing HeLa cells, we visualized the uneven distribution of Venus excitation and

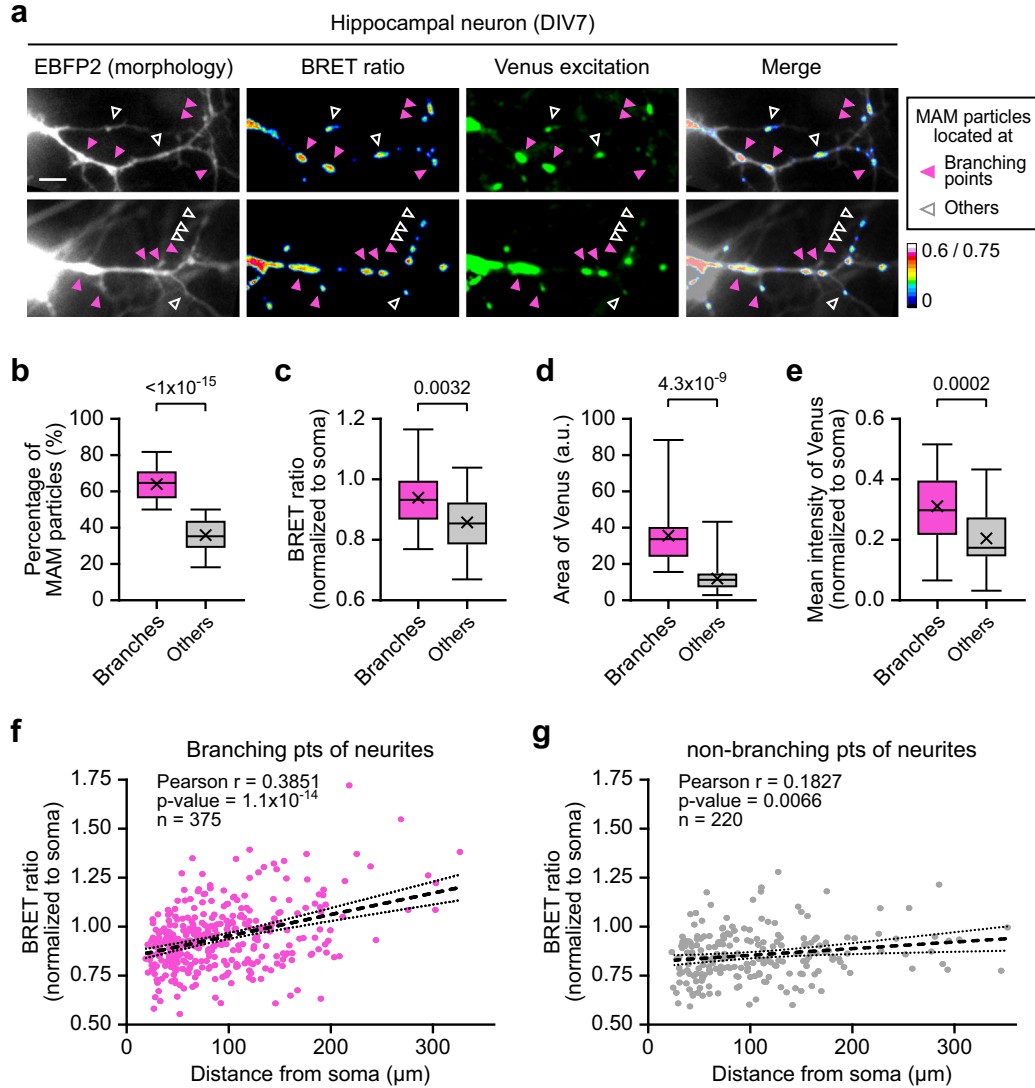

**Fig. 4 | MAM-Calflux can determine the local distribution of MAM-specific calcium dynamics. a–e** MAM-Calflux visualized the MAM puncta enriched at the branching points of developing neurites with higher $Ca^{2+}$ levels. Representative images (**a**) for BRET signals upon the MAM puncta at the neurites of DIV7 hippocampal neurons. The magenta-filled arrowheads indicate MAM puncta at the branching points, and the empty white arrowheads indicate MAM puncta at the non-branching points. The fraction of MAM puncta at branching points in each neuron (**b**) was higher than the non-branching point. BRET ratio (**c**), MAM area (**d**), and mean Venus excitation intensities (**e**) of MAM puncta at the branching points were higher than others. (*n* = 27 neurons). **f–g** Correlation between BRET signals and distance from the center of soma among the MAM puncta at the branching points (**f**) and non-branching points (**g**). The scale bar represents 10 μm. All results are presented as box plots representing the median and interquartile range with whiskers min/max value and the cross representing the mean value. All *P* values were calculated using two-tailed Student's *t* test for **b–e** and Pearson correlation coefficient analysis with two-tailed *P* values for **f, g**. For **f, g**, each dot represents each MAM puncta (*n* = 375 for branching points (**f**) and *n* = 220 for non-branching points (**g**) from 27 individual neurons). For **f, g**, black dashed lines represent simple linear regression with black dotted lines representing 95% confidence intervals. Source data from **b–g** are provided as a Source Data file.

BRET ratio at the steady-state (pre-treatment) (Supplementary Fig. 8c). Notably, after histamine treatment, we observed that the BRET ratio was unevenly increased, and some MAM regions responded more strongly than others (Supplementary Fig. 8d), as previously suggested[37]. In contrast, the Venus excitation pattern was not largely affected by histamine, indicating that the uneven increase in the BRET signal was not induced by MAM structural changes (Supplementary Fig. 8e). Moreover, the direct $Ca^{2+}$ influx due to $CaCl_2$ increased $Ca^{2+}$ signals in a general manner (Supplementary Fig. 8f, g), implying that the uneven increase of the BRET signal due to histamine could be attributed to the different IP3R-mediated $Ca^{2+}$ responses between MAM regions.

Next, we analyzed MAM $Ca^{2+}$ distribution in neuronal processes in DIV7 hippocampal neurons. We quantitated BRET signals in each MAM puncta and found a positive correlation between the BRET ratio and the distance from the soma (Supplementary Fig. 9). Moreover, approximately two-thirds of the MAM puncta in neurites were located at the branching points (Fig. 4a, b) and exhibited relatively higher $Ca^{2+}$ levels and larger MAM areas (Fig. 4c–e). When we separately analyzed the correlation between the BRET ratio and the distance from the soma, MAM puncta at branching points had a stronger positive correlation than the ones at non-branching points (Fig. 4f, g). These results support that MAM accumulates at neurite branching points with pronounced $Ca^{2+}$ dynamics that may be necessary properties such as local $Ca^{2+}$ buffering[42,43]. These findings support the potential of MAM-Calflux in understanding the intracellular distribution of MAM structure and MAM $Ca^{2+}$ dynamics in various cellular processes.

## MAM-Calflux can monitor altered MAM-calcium physiology in neurodegenerative disease model neurons

Recent studies suggest that neurodegenerative diseases, especially PD, AD, and ALS, are closely related to abnormal regulation of the MAM structure and thereby its functions. However, a deeper understanding of MAM $Ca^{2+}$ physiology in these diseases is still required[44]. We introduced MAM-Calflux into the neurons of mouse models pertaining to these diseases to quantitate steady-state $Ca^{2+}$ levels inside the MAM region, which was not previously achievable.

5xFAD is an AD mouse model that expresses disease-linked mutant forms of human APP and PSEN1[45]. At 7 days in vitro (DIV), the primary cultured 5xFAD hippocampal neurons had a decreased BRET ratio and Venus excitation intensities, indicating that both MAM $Ca^{2+}$ levels and MAM extent were affected (Fig. 5a–d). Moreover, MAM $Ca^{2+}$ levels were reverted in those neurons by expressing the VAPB-PTPIP51 complex, thereby reverting MAM structure, indicating that decreased steady-state MAM $Ca^{2+}$ levels in part arising from loosened MAM (Fig. 5e–i). PINK1 (PTEN-induced kinase 1) is a serine/threonine kinase responsible for mitochondrial quality control, and its loss-of-function mutations cause autosomal recessive early-onset PD[46]; accumulating evidence indicates an association between PINK1 and MAM biology[47,48]. PINK1 KO neurons had decreased MAM $Ca^{2+}$ concentrations, whereas MAM structure was not significantly altered (Supplementary Fig. 10a–d). *SOD1* (superoxide dismutase 1) is the second most frequent gene associated with familial ALS, and the accumulation of mutant SOD1 proteins in the MAM has been reported[17]. SOD1*G93A transgenic mouse neurons, which ectopically express human SOD1 with ALS-linked mutation, had a minimal effect on the $Ca^{2+}$ dynamics and MAM structure (Supplementary Fig. 10e–h).

Alpha-synuclein (SNCA or α-syn) protein, which is located at the presynapse and regulates synaptic vesicles, is one of the most prominent causes of PD, as evidenced by its presence in Lewy bodies and missense mutations discovered in familial PD[49]. Ectopic expression of disease-linked SNCA mutants or excessive delivery of SNCA proteins has been reported to reduce ER-mitochondria contact and $Ca^{2+}$ transfer[12,50]. When we examined MAM-Calflux in SNCA*A53T transgenic mouse neurons, which over-express the PD-linked A53T mutant form of human SNCA[51,52], MAM steady-state $Ca^{2+}$ levels were significantly enhanced (Fig. 6a, b). In contrast, the intensity of MAM structure was diminished (Fig. 6c, d), which is consistent with previous reports. BRET signals were not reverted by expressing the VAPB-PTPIP51 complex. However, it recovered Venus signals, indicating that increased steady-state MAM $Ca^{2+}$ levels were largely independent of the loosened MAM structure (Fig. 6e–i). These results indicate that excessive α-syn proteins reduce the MAM structural extent, resulting in abnormal $Ca^{2+}$ accumulation in MAM regions with possible link to PD pathogenesis.

## Discussion

Although $Ca^{2+}$ transfer was assumed to be one of the major functions of MAM, specific and direct measurement of $Ca^{2+}$ concentration in the MAM was not possible due to technical limitations. Instead, MAM $Ca^{2+}$ has been evaluated by measuring cytosolic $Ca^{2+}$ levels near the OMM[14,36] or by comparing $Ca^{2+}$ dynamics in the ER and mitochondria after stimulation of $Ca^{2+}$ crosstalk[6,31]. Csordas et al. presented an advanced methodology utilizing rapamycin-inducible bridge-forming modules (RiBFM) to guide OMM-targeted Pericam toward the ER; thus, the signal was more specific to MAM, although this still could not exclude non-MAM $Ca^{2+}$ signals and had a chance of artificial increase of MAM structures[53]. In this study, we modified the fluorescent domain of the ratiometric calcium indicator to be specifically located inside the MAM, which could directly and specifically measure steady-state MAM $Ca^{2+}$ levels without external stimuli. Its ratiometric character enables the estimation of MAM $Ca^{2+}$ physiology with cell-to-cell or region-to-region comparisons. For instance, when we compared steady-state BRET ratios of different cell types which were almost simultaneously

evaluated, such as HeLa control groups and primary cultured hippocampal neuron control groups in Fig. 3a–d, Supplementary Fig. 5a–d, and Supplementary Fig. 7, the neuronal cells exhibited higher MAM $Ca^{2+}$ signals than that of HeLa cells, although the MAM structure intensities (Venus excitation) were similar. This implies that MAM in neuronal cells contains more activated $Ca^{2+}$ transfer machinery or a higher frequency of MAM $Ca^{2+}$ crosstalk. Therefore, the availability of data on the MAM-Calflux BRET ratio values corresponding to $Ca^{2+}$ concentrations is critical for the precise measurement of MAM $Ca^{2+}$ concentrations. Further applications of MAM-Calflux, such as comparisons of MAM $Ca^{2+}$ concentrations in various cellular compartments and in response to various extracellular conditions, would help comprehend MAM $Ca^{2+}$ physiology.

The specific characteristics of the BRET-based calcium indicator coupled with the BiFC strategy have imparted the dual functionality in MAM-Calflux as a MAM $Ca^{2+}$ indicator and MAM structural marker (Fig. 1c). While given the substrate, the bioluminescence signal and BRET-based Venus signal indicate MAM $Ca^{2+}$ concentrations. Independent of the substrate, laser excitation-based fluorescent imaging of Venus specifically exhibits the MAM structure since the Venus BiFC fragments can be fused only at MAM. Moreover, we verified that the detected BRET-based Venus signal was negligible compared with that of the excitation-based Venus fluorescence signal; thus, the MAM $Ca^{2+}$ level and MAM structure could be simultaneously analyzed even during time-lapse imaging. Taking advantage of this dual functionality, we confirmed the previously reported local distribution of MAM $Ca^{2+}$ physiology[36,53]. Under basal conditions, the MAM $Ca^{2+}$ level was distinct from the MAM marker intensity, especially in polarized cells such as neurons (Fig. 4 and Supplementary Fig. 8a, b). On the one hand, MAM puncta stalled at the neurite branching points of developing neurons with higher $Ca^{2+}$ levels than those of nearby MAMs. Emerging evidence emphasizes the importance of local $Ca^{2+}$ modulation for proper axon/dendrite arborization[54] and the positioning of organelles and cytoskeletons at the branching points[42,43]. Therefore, our findings are likely to provide insights into the roles of the local MAM-mediated $Ca^{2+}$ dynamics in neuronal process development. We also observed that some MAM regions exhibited particularly strong $Ca^{2+}$ signals during histamine-stimulated ER $Ca^{2+}$ release by IP3Rs (Supplementary Fig. 8c–e). This possibly supports previous findings that IP3R clusters exist for local retuning $Ca^{2+}$ activity and that IP3R isoforms differentially regulate MAM and local $Ca^{2+}$ transfer[55,56]. Therefore, we suggest that further investigation using MAM-Calflux can elucidate the local MAM $Ca^{2+}$ physiology under various conditions independent of local MAM structure formation.

In addition, the dual functionality of MAM-Calflux could help distinguish whether the change in MAM $Ca^{2+}$ concentration is caused by the MAM structural change or by other factors such as altered $Ca^{2+}$ transporter activity. Since the MAM provides the physical site for $Ca^{2+}$ crosstalk between the ER and mitochondria, it is plausible to expect that a tighter MAM structure resulting in greater MAM $Ca^{2+}$ dynamics. When the MAM structure was enhanced by STS treatment or ectopic expression of tethering proteins, MAM-Calflux demonstrated an increase in the steady-state MAM $Ca^{2+}$ level (Supplementary Fig. 7 and Fig. 3). In contrast, alterations in the $Ca^{2+}$ machinery affect the MAM $Ca^{2+}$ dynamics without affecting the MAM structure. For instance, the knockout of B-cell lymphoma-extra large (Bcl-xL), an activator of voltage-dependent anion-selective channel protein 1 (VDAC1), decreased IP3R-mediated ER-mitochondria $Ca^{2+}$ transfer. In contrast, the MAM structure was not affected[57]. We introduced MAM-Calflux into SNCA*A53T mouse neurons, indicating that the steady-state MAM $Ca^{2+}$ level was significantly higher than that of the wild-type, whereas the MAM structural intensity decreased (Fig. 6). These upregulated $Ca^{2+}$ levels were not affected by VAPB-PTPIP51 expression, indicating that this is independent of the loosened MAM structure. Indeed, α-syn was reported to interact with and block VDAC1, reducing $Ca^{2+}$ influx to

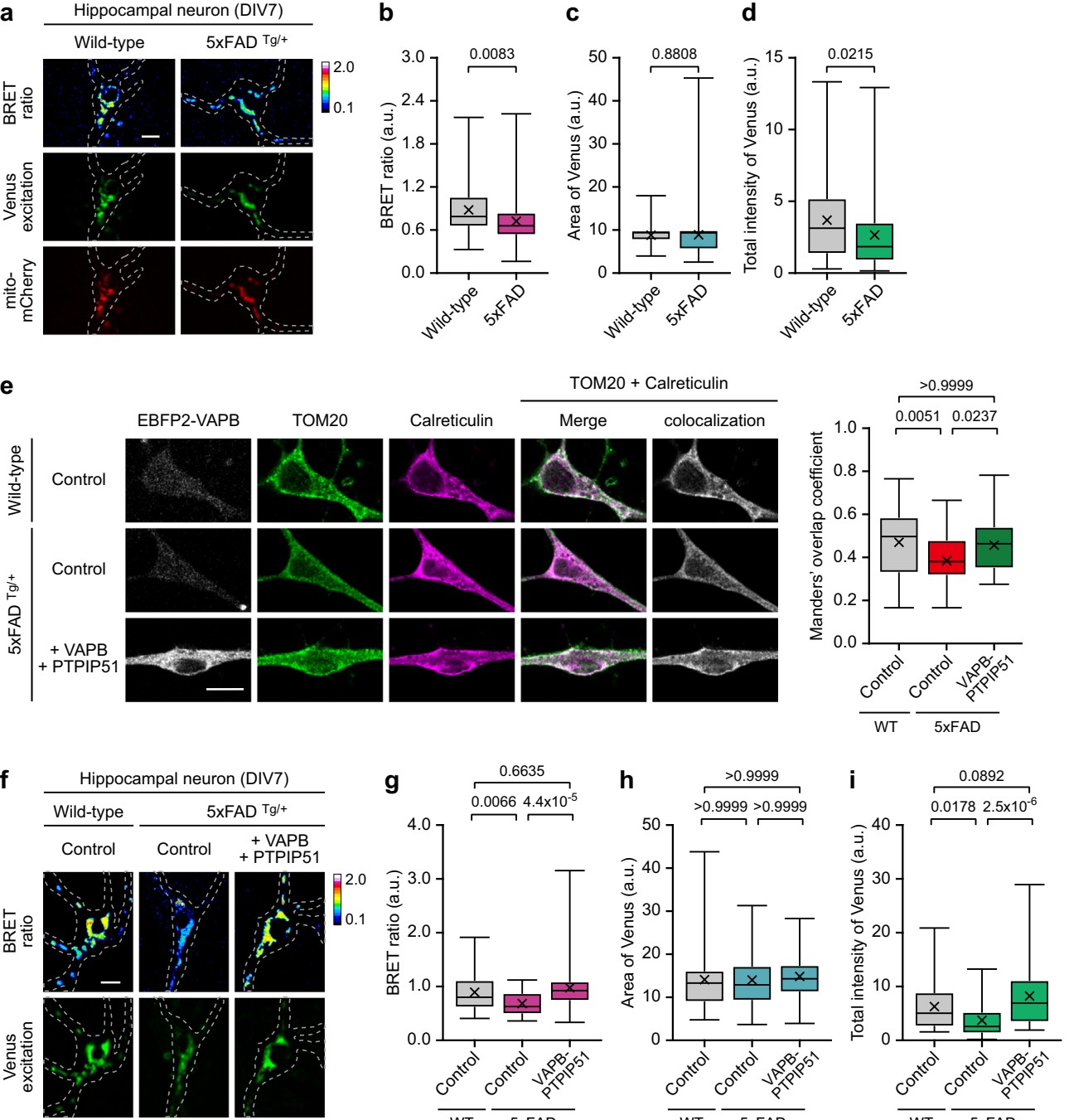

**Fig. 5 | MAM-Calflux can monitor altered MAM structure and MAM Ca²⁺ levels in Alzheimer's disease model neurons. a–d** MAM-Calflux detected the decreased MAM Ca²⁺ levels and MAM integrity in the soma of primary cultured 5xFAD tg mouse neurons. Representative images (**a**) and bar graphs representing BRET ratio (**b**), MAM area (**c**), and total Venus excitation intensities (**d**). (*n* = 70 for wild-type neurons and *n* = 76 for 5xFAD tg neurons). **e** Representative images (left) and Manders' overlap coefficients (right) of colocalization between mitochondria (TOM20) and ER (Calreticulin) among 5xFAD tg neurons with VAPB-PTPIP51 tethering complex over-expression. (*n* = 50 for wild-type neurons, *n* = 43 for 5xFAD tg neurons, and *n* = 48 for VAPB-PTPIP51 expressing 5xFAD tg neurons).

**f–i** Representative images (**f**) and graphs representing BRET ratio (**g**), MAM area (**h**), and total Venus excitation intensities (**i**). (*n* = 48 for wild-type neurons, *n* = 54 for 5xFAD tg neurons, and *n* = 54 for VAPB-PTPIP51 expressing 5xFAD tg neurons). Dashed lines in **a** and **f** represent neuronal morphologies. The scale bars represent 10 μm. All results are presented as box plots representing the median and ianterquartile range with whiskers min/max value and the cross representing the mean value. All *P* values were calculated using two-tailed Student's *t* test for **b–d** and one-way ANOVA with Bonferroni's multiple comparison tests for **e** and **g–i**. Source data from **b–e** and **h–i** are provided as a Source Data file.

mitochondria and leading to the accumulation of Ca²⁺ on the MAM side[58]. In addition, α-syn aggregation binds to sarco/endoplasmic reticulum Ca²⁺-ATPase (SERCA) and increases cytosolic Ca²⁺ level at a later stage[59]. Therefore, based on results from MAM-Calflux, we hypothesize that the pathogenic α-syn decreases Ca²⁺ uptake by

mitochondria, resulting in the abnormal accumulation of Ca²⁺ in the MAM regardless of its loosened structure. Likewise, defining both MAM structure and MAM Ca²⁺ by the dual functionality of MAM-Calflux can provide more insights into understanding MAM Ca²⁺ physiology in various cellular or disease-relevant conditions.

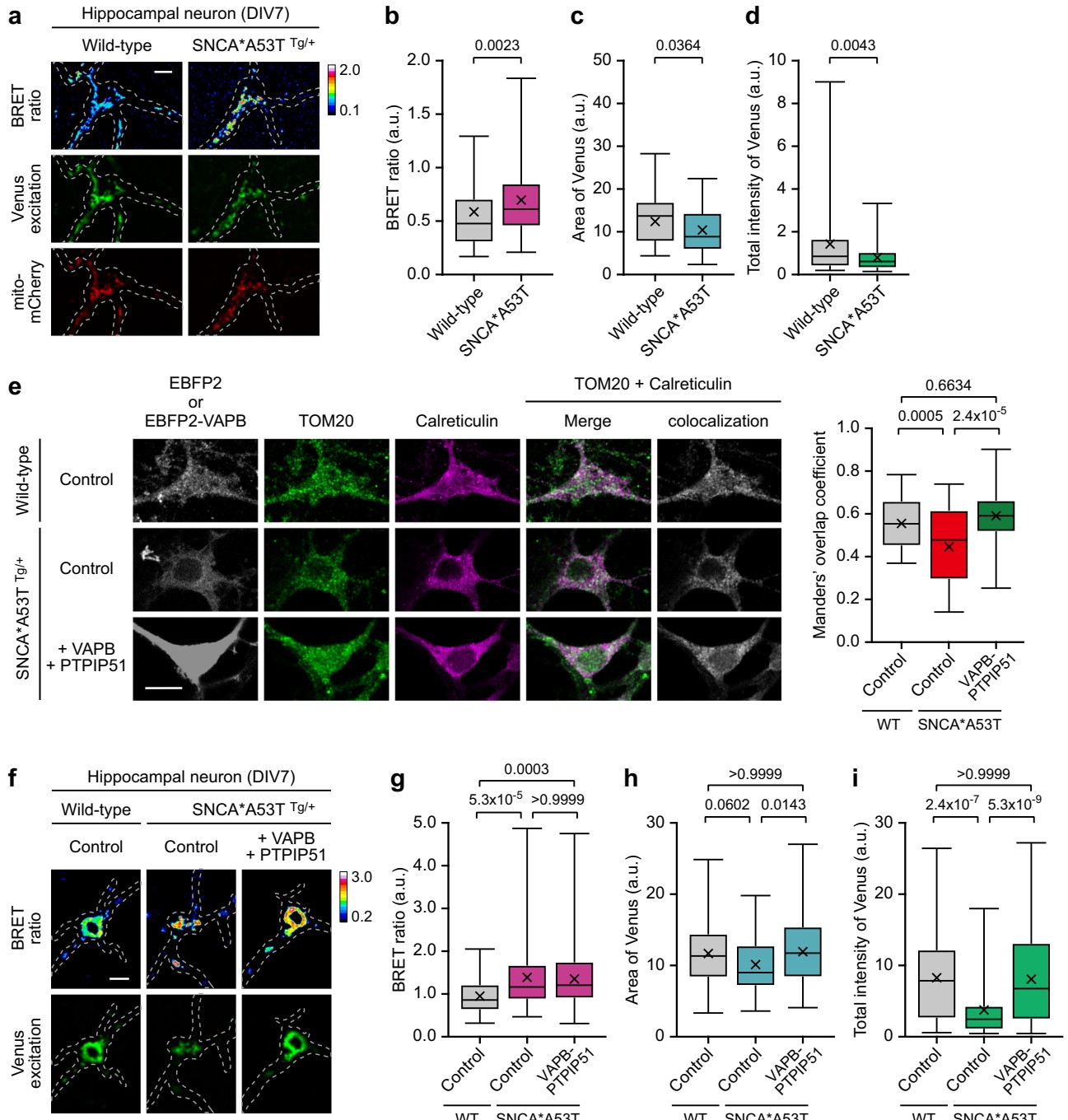

**Fig. 6 | MAM-Calflux can determine the altered MAM-calcium physiology in the α-syn-mediated Parkinson's disease model neurons. a–d** MAM-Calflux detected the increased MAM steady-state $Ca^{2+}$ levels with diminished structure integrity in primary cultured SNCA*A53T tg mouse neurons at DIV7. Representative images (**a**) and bar graphs representing BRET ratio (**b**), MAM area (**c**), and total Venus excitation intensities (**d**). ($n = 71$ for wild-type neurons and $n = 66$ for SNCA*A53T tg neurons). **e** Representative images (left) and Manders' overlap coefficients (right) of colocalization between mitochondria (TOM20) and ER (Calreticulin) among SNCA*A53T tg neurons with VAPB-PTPIP51 tethering complex over-expression. ($n = 55$ for wild-type neurons, $n = 43$ for SNCA*A53T tg neurons, and $n = 36$ for VAPB-PTPIP51 expressing SNCA*A53T tg neurons). **f–i** Representative images (**f**)

and graphs representing BRET ratio (**g**), MAM area (**h**), and total Venus excitation intensities (**i**). Dashed lines in **f** represent the neuronal morphologies. ($n = 77$ for wild-type neurons, $n = 95$ for SNCA*A53T tg neurons, and $n = 92$ for VAPB-PTPIP51 expressing SNCA*A53T tg neurons). Dashed lines in **a** and **f** represent the neuronal morphologies. The scale bars represent 10 μm. All results are presented as box plots representing the median and interquartile range with whiskers min/max value and the cross representing the mean value. All *P* values were calculated using two-tailed Student's *t* test for **b–d** and one-way ANOVA with Bonferroni's multiple comparison tests for **e** and **g–i**. Source data from **b–e** and **g–i** are provided as a Source Data file.

The MAM-specific localization and sustained $Ca^{2+}$ response of MAM-Calflux proves that this strategy can be employed in developing calcium indicators specific for other inter-organelle contacts by making appropriate substitutions for organelle-specific localization signals

of MAM-Calflux and optimizing the linker lengths[60]. For instance, the plasma membrane (PM)-ER contact serves for the efficient transfer of extracellular $Ca^{2+}$ to the ER lumen and activates ER $Ca^{2+}$ release under specific conditions such as muscular contraction[61]. Thus, interrogating

Ca²⁺ crosstalk between the PM and ER would be informative. Dolman et al. determined that the mitochondria may be responsible for the Ca²⁺ gradient in *trans*- and *cis*-Golgi by close contact with Golgi apparatus[62]. Since Ca²⁺ dynamics in the Golgi affect its structures and functions, especially secretory pathways and local Ca²⁺ buffering[63], the interaction between mitochondria and the Golgi apparatus is another possible candidate to examine Ca²⁺ crosstalk. Calcium indicators specific to these inter-organelle structures may advance our understanding of organellar Ca²⁺ dynamics and organelle-to-organelle Ca²⁺ crosstalk within the cell.

Although the MAM-Calflux is a versatile tool for studying MAM-specific Ca²⁺ physiology, a number of further improvements can be made. First, the split fragments of Venus by BiFC have been reported to undergo spontaneous self-assembly, although this was reduced when each fragment was fused to other proteins[64,65]. Self-assembly of Venus in MAM-Calflux would ectopically enhance MAM structure formation, resulting in the over-estimation of MAM Ca²⁺ basal levels. Indeed, we observed that expression of MAM-Calflux for long periods can affect MAM structure formation, requiring careful design of experimental conditions, such as expression period and levels. This may be partially overcome by modifying the strategy for Venus fragmentation in MAM-Calflux, for instance, introducing mutations in Venus[66,67], changing Venus split residues[68], and utilizing dimerization-dependent fluorescent protein[69]. Employing low-expression promoters or inducible expression systems will also be beneficial. Also, a relatively low bioluminescence signal and BRET-based Venus signal can restrict the frame-to-frame interval of time-lapse imaging. In our imaging condition, we achieved the shortest interval of 5 sec to observe MAM Ca²⁺ dynamics against external stimuli. Still, this interval may be insufficient to precisely analyze Ca²⁺ dynamics in some conditions, indicating room for further improvement in conjunction with the conventional BRET system[27].

## Methods

### Ethical Statement
All animal procedures were approved by the Institutional Animal Care and Use Committee (IACUC) of Pohang University of Science and Technology (POSTECH-2022-0085). All experiments were performed in accordance with the approved guidelines.

### Animals
Pregnant B6 mice used for primary hippocampal neuron culture were purchased from Hyochang Science (Daegu, South Korea). SNCA*A53T mice were a generous gift from Dr. Won-Jong Oh (Korea Brain Research Institute)[52]. PINK1-deficient mice were a generous gift from Dr. Xiaoxi Zhuang (Chicago University) and Dr. Sang Myun Park (Ajou University School of Medicine)[70]. 5xFAD mice were a generous gift from Dr. Kyong-Tai Kim (Pohang University of Science and Technology)[45]. SOD1*G93A mice[71] were purchased from The Jackson Laboratory (Stock No. 002726) and maintained on a C57BL/6 background. The animals were group-housed under diurnal light conditions (12 hr light, 12 hr dark cycle) and had free access to food and water. (temperature 22 °C ± 2 °C, humidity 50% ± 5%).

### Antibodies and plasmids
Anti-TOM20 mouse monoclonal antibodies (Cat# ab56783, Abcam, and Cat# sc-17764, Santa Cruz Biotechnology), anti-Calnexin (Cat# ab22595, Abcam), and anti-Calreticulin (Cat# ab2907, Abcam) were used for immunocytochemistry experiments for 1:50–1:200. For immunostaining, Alexa Fluor 568 or 647 conjugated goat anti-mouse IgG (Cat# A-11004 and Cat# A-21236, Molecular Probes) and Alexa Fluor 647 conjugated goat anti-rabbit IgG (Cat# A-21244, Molecular Probes) were used as secondary antibodies for 1:200.

pT7-CalfluxVTN, which was a gift from Carl Johnson (Addgene plasmid # 83926)[27], was served as the template to design MAM-Calflux. A 173 aa N-terminal fragment of Venus domain of CalfluxVTN, VN173, was conjugated with a 103 aa linker and the ER-targeting sequence (mSac1 521–587 aa) at its C-terminus. A fragment containing the C-terminal part of Venus, VC155, and following Troponin C and NanoLuc domains was conjugated with the mitochondria-targeting sequence (mAkap1 1–30 aa) and a 57 aa linker at its N-terminus. Sec61b-mCherry was a gift from Gia Voeltz (Addgene plasmid # 49155)[72]. pCMV R-CEPIA1er was a gift from Masamitsu Iino (Addgene plasmid # 58216)[29]. The core sequence of human MFN2 shRNA was GGAAGAGCACCGTGATCAATG[73].

### Cell culture and transfection
HeLa (ATCC # CCL-2) cells were grown in Dulbecco's Modified Eagle Medium (DMEM) (HyClone) supplemented with 10% (v/v) fetal bovine serum (FBS) (Gibco) and 1% penicillin/streptomycin (Gibco). The cell-line was authenticated using STR profiling method and tested negative for mycoplasma contamination. For imaging samples, ~1.5 × 10⁵ cells/mL of cells were seeded on the 18-mm or 25-mm cover glass in 12-well or six-well plates, respectively. The cells were transfected using transfection reagents, either polyethylenimine (PEI) or Lipofectamine 2000 (Thermo Fisher Scientific). For transfection in a 12-well plate, 300–400 ng of MAM-Calflux-P2A construct and each 400-600 ng of other plasmids, such as EBFP2-N1, EBFP2-hVAPB, FLAG-hPTPIP51, MYC-hGRP75, and pLL3.7-mRFP-MFN2 shRNA, were mixed in 100 μL Opti-MEM with 4 μL PEI or 1 μL Lipofectamine 2000.

Primary cultures of hippocampal neurons were established by isolating E16 B6 mouse embryo hippocampal tissues in Hanks' balanced salt solution (HBSS) (Gibco) and dissociating the tissues in 0.25% trypsin (Sigma-Aldrich) and 0.1% DNase I (Sigma-Aldrich) for 10 min at 37°C. Cells were resuspended in Neurobasal medium (Gibco) supplemented with 10 mM HEPES (pH 7.4) and 10% (v/v) horse serum to a final cell concentration of 4.0 × 10⁵ cells/mL. The cells were plated on glass coverslips pre-coated with poly-D-lysine and laminin. After 2 h of plating, the cell medium was replaced with Neurobasal medium containing 2 mM glutamine, 2% (v/v) B27 supplement (Gibco), and 1% (v/v) penicillin/streptomycin. The neurons were transfected with Lipofectamine 2000, and the medium was replaced with the culture medium 2 h after transfection. For transfection in a 12-well plate, 500–600 ng of MAM-Calflux-P2A construct and each 800–1000 ng of other plasmids were mixed in 100 μL Opti-MEM with 2 μL Lipofectamine 2000.

### Immunocytochemistry
For immunocytochemistry, the cells were fixed with 4% paraformaldehyde in PBS or 4% paraformaldehyde and 4% sucrose in PBS for 20 min and washed with PBS for three times. Cells were permeabilized with 0.2% Triton-X-100 in PBS for 5 min and blocked with 5% goat serum or 5% BSA in PBS for >30 min. For protein staining, the cells were incubated with primary antibodies diluted in the blocking solution for 1 h at room temperature or overnight at 4°C, washed with PBS for three times, and treated with secondary antibodies diluted in the blocking solution for 1 h at room temperature. Cell images were acquired using FV3000 confocal laser scanning microscope (Olympus) and processed using cellSens 2.3 software (Olympus) and ImageJ 1.52p (Fiji) software (National Institute of Health)[74]. Images were deconvolved using the advanced constrained iterative (CI) algorithm-based deconvolution program of cellSens software. Colocalization between ER and mitochondrial markers was quantitated using cellSens software, Imaris 9.2 software (Bitplane), or Fiji software.

### BRET measurement using plate reader system
To obtain in vitro purified Calflux proteins, the His6-tagged split-version of CalfluxVTN (lacking targeting sequences and linkers in MAM-

Calflux) was introduced to the BL21 strain and cultured at 37 °C in LB medium containing 100 mg/mL ampicillin, until an $OD_{600nm}$ of 0.6 was achieved. Subsequently, a 16-hour induction was performed using 1 mM isopropyl β-D-1-thiogalactopyranoside (IPTG) at 16 °C, followed by cell harvesting via centrifugation and snap-freezing in liquid nitrogen. The resulting cell pellets were resuspended in lysis buffer (20 mM Tris pH 8.0, 100 mM NaCl, 2 mM EDTA, 1% Triton-X, protease inhibitor, 1 mg/ml lysozyme, pH 7.4), incubated for 30 min at 37 °C, and sonicated (5 min at 20% amplitude, 5 s on, 2 s off). Cellular debris was removed through centrifugation at 7000 × g for 15 min at 4 °C, and the resulting clear supernatant was combined with Ni-NTA agarose beads in phosphate-buffered saline (PBS), followed by a 4-hour incubation at 4 °C. The beads were subsequently washed with a washing buffer (20 mM Tris pH 8.0, 100 mM NaCl, 20 mM imidazole), and the His-tagged proteins were eluted using an elution buffer (20 mM Tris pH 8.0, 100 mM NaCl, 300 mM imidazole). Finally, the proteins were dialyzed in a dialysis buffer (20 mM Tris-HCl pH 8.0, 100 mM NaCl, 3% glycerol, and 1 mM DTT). In vitro purified Calflux-N term and Calflux-C term were mixed to calcium calibration buffer being 1 μM and 0.05 μM for final concentrations, respectively. Calcium calibration buffer was made by mix of Ca-EGTA (10 mM CaEGTA, 100 mM KCl, and 30 mM MOPS pH 7.2) and zero calcium buffer (10 mM $K_2$EGTA, 100 mM KCl, and 30 mM MOPS pH 7.2).

For live cell samples, HeLa cells in six-well plate were transfected with MAM-Calflux-C or MAM-Calflux. After 6 h, the cells were detached using trypsin-EDTA, resuspended in complete DMEM, and transferred to a white-bottom 96-well plate. For emission wavelength scan, after 12-16 h, the medium was replaced with $Ca^{2+}$-free HBSS (Cat# 14175-095, Gibco) with 2 μM ionomycin and either 5 mM EGTA or variable concentrations of $CaCl_2$ from 0.001 to 100 mM for 10−15 min.

Luciferase was activated by treating the cells with 50 μM of furimazine (Cat# AOB36539, Aobious) substrate. After stabilization for 2 minutes, the emission intensities of wavelengths from 400 to 600 nm with 2 nm intervals were measured at 37 °C using a multifunctional microplate reader (Tecan Infinite M200 pro).

## BRET imaging using EMCCD equipped fluorescence microscope

HeLa or primary cultured neurons on the cover glass were transfected with MAM-Calflux and EBFP2-N1, EBFP2-hVAPB and FLAG-hPTPIP51, and Myc-hGRP75. For MFN2 knockdown, HeLa cells were transfected with pLL3.7-mRFP-MFN2 shRNA, moved to a cover glass after 24−30 h of transfection, and again transfected with MAM-Calflux after 24−30 h of seeding. After 18−24 h, the cover glass was transferred to a live imaging chamber, and the cell medium was replaced with $Ca^{2+}$-free HBSS with 10 mM HEPES and 50 μM furimazine. To maintain a sufficient concentration of furimazine, at least 250 μL of buffer was used for the 18-mm coverslip. After 5 min of stabilization with furimazine, BRET imaging was performed with an inverted fluorescence microscope (IX71, Olympus) using ×20 (UApoN340 ×20/0.70 W infinite/0.17/FN22), ×40 (UPlanSApo ×40/0.95 infinite/0.11-0.23/FN26.5), and ×60 (UPlanSApo ×60/1.35 Oil infinite/0.17/FN26.5) objective lenses at 37 °C. To get a 525/460 nm ratio image, a dual-view beam splitter (Photometrics) was installed in front of the EMCCD camera (Cat# C9100-13, Hamamatsu). To obtain the BFP/RFP and Venus images separately, a triple dichroic mirror (59001bs, Chroma) with dual-view emission filters for 525 nm (AT525/30 m, Cat# 223164, Chroma) and for 460 nm and 600 nm (59003 m, Cat# 365631, Chroma) were used. Images were acquired using MetaMorph 7.7 software (Molecular Devices) with multi-dimensional acquisition for the Em525 and Em460 dual image, Venus excitation image, and BFP excitation image, sequentially. Each image was obtained at 690 kHz, 11×, gain 100, binning 1, 10 s for Em460/Em525 imaging and 690 kHz, 1×, 500 ms for Venus excitation imaging. These conditions prevented the crosstalk of signals from Em525 into Venus excitation images. The BFP excitation imaging was obtained at 2.75 MHz, 1×, 300 ms. The images were analyzed using Fiji with the use of "Template Matching" plug-in to align Em460 and Em525 images, "Subtract Background" function or "BRET Analyzer 1.0.7" plug-in to subtract background signals, and "BRET Analyzer 1.0.7" plug-in to produce BRET ratio images. To calculate mean BRET ratio after masking MAM regions, "Threshold" and "Analyze Particles" functions were used over Venus excitation images to determine MAM-specific region-of-interests (ROIs).

## Live $Ca^{2+}$ imaging using MAM-Calflux or mitochondria-, cytosolic-, and ER-specific $Ca^{2+}$ sensors

Live calcium imaging was performed as previously described, with some modifications[6,75]. To measure the $Ca^{2+}$ responses of subcellular organelles, HeLa cells or primary cultured hippocampal neurons were transfected with MAM-Calflux and EBFP2-N1, cyto-GCaMP6s and R-CEPIA1er, or mito-GCaMP6s and R-CEPIA1er constructs on DIV6. The cells were loaded with $Ca^{2+}$-free HBSS and sequentially exposed to 10 μM histamine and 10 μM $Ca^{2+}$-free ionomycin (Cat# I9657, Sigma-Aldrich) with 1 mM $CaCl_2$. For a single treatment of histamine, cells were loaded with HBSS containing 1 mM $MgCl_2$ and 2 mM $CaCl_2$. To maintain a sufficient concentration of furimazine during time-lapse imaging, at least 250 μL of buffer was used for the 18-mm coverslip, and images were acquired within 20−25 min.

BRET images were recorded in MetaMorph software with multi-dimensional acquisition and time-lapse functions for the Em525 and Em460 dual image, the Venus excitation image, and the BFP excitation image, sequentially. In total 25−35 frames were acquired, and imaging conditions and interval times were adjusted to finally obtain ~15 seconds between frames. After image processing, the amplitude ($\Delta B/B_0$) of each neuron was calculated as $(B-B_0)/B_0$, where $B_0$ is the baseline BRET ratio signal averaged over five frames before stimulation and B is the peak intensity of the BRET ratio in the response.

The fluorescence intensities were recorded at 37 °C with supplying 5% $CO_2$ gas in FV31S-DT software at intervals of 2 seconds for 200 seconds in total using FV3000 confocal laser scanning microscope (Olympus) with UPLSAPO ×20/0.75 NA objective. Background fluorescence was subtracted and the amplitude ($\Delta F/F_0$) of each neuron was calculated as $(F-F_0)/F_0$, where $F_0$ is the baseline mito-GCaMP6s, cyto-GCaMP6s, or R-CEPIA1er fluorescence signal averaged over 20 seconds before the stimulation and $F$ is the peak intensity of fluorescence in the response.

## Statistics and reproducibility

All graphs are presented as the mean ± SEM or box plots representing the median and interquartile range with whiskers min/max value and the cross representing the mean value. The statistical significance of the data was analyzed using two-tailed Student's $t$ test for comparisons between two groups and one-way ANOVA followed by Bonferroni's post-hoc test for comparisons among multiple groups. All statistical analysis, correlation analysis, and fitting with a sigmoidal curve model or a simple linear regression model were calculated using Prism 9.4 (GraphPad).

Sample sizes for all statistical evaluations are indicated in Figure legends. No statistical method was used to predetermine sample size. Number of cells per group was chosen based on sample size in the calcium measurement experiment with microscopic images which meets or exceed previous studies[6,60]. No data were excluded from the analyses. All sample images were acquired in random order within the sets and all cells were chosen at the random site. Experimenters were blinded to each group allocation during the image analysis by removing any sample information in file names.

## Reporting summary

Further information on research design is available in the Nature Portfolio Reporting Summary linked to this article.

## Data availability

All data are available in the main text or the supplementary materials. Source data for all figures are provided with the paper. Materials are available from the corresponding author. Source data are provided with this paper.

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

## Acknowledgements

This work was supported by funds from a Samsung Science and Technology Foundation grant (SSTF-BA1601-14) and the National Research Foundation of Korea (NRF-2020M3E5E2039894, NRF-2021R1A2C3010639, and NRF-2017R1A5A1015366 to S.K.P.). This study was also supported by KBRI Basic Research Program funded by the Korean Ministry of Science and ICT (21-BR-03-01 to S.K.P.) and Korea Initiative for Fostering University of Research and Innovation Program (NRF-2020M3H1A1075314 to Y.W.).

## Author contributions

E.C., Y.W., Y.S., and S.K.P. conceptualized the study. E.C., Y.W., B.K.S., S.J.K., and T.T.M.N. performed the experiments and analyzed the data. E.C., Y.W., Y.S., B.K.S., T.T.M.N., J.Y.Y., T.D.N., S.B.L., D.J.M., and S.K.P. participated in designing the methodology. E.C., Y.W., Y.S., and S.K.P. provided experimental resources and assisted with the final preparation of the manuscript. S.K.P. supervised the study. E.C., Y.W., and S.K.P. wrote the manuscript. All authors read and approved the final version of the manuscript.

## Competing interests

The authors declare no competing interests.
