## [Peer Review File · Nature Communications]

REVIEWER COMMENTS

Reviewer #1 (Remarks to the Author):

Review of NatComm paper.

Inter-organellar contacts between ER and mitochondria (MAM) are considered of importance for a host of cellular functions. They govern the calcium (Ca^{2+}) flux between ER and mitochondria, that is driven by the high Ca^{2+} concentration in ER and regulated by the IP3R channel in ER, the VDAC channel in mitochondria and the assembly complex between the organelles. The MAM are dynamic structures that can be assembled and disassembled at a fast pace, and their activity, e.i. the flux of Ca^{2+} difficult to measure.

The manuscript by Cho et al., describes the creation of a novel sensor based on the bimolecular complementation of a split fluorescent Venus protein with one half attached to a ER localization segment from Sac1 and the other half comprising a mitochondrial localization segment from AKAP, along with a Ca^{2+} -binding domain from troponin C and a luminescent luciferase construct.

This assembly aims to tackle two questions. First, the localization of MAM to be visualized by fluorescence upon excitation of the complemented Venus protein, that can assemble when ER and mitochondrial membranes are in close proximity. Second, to measure the Ca^{2+} level in close proximity to the MAM, the binding of Ca^{2+} to the troponin C domain changes its conformation whereby the NanoLuc domain is brought into proximity of the Venus protein. When furimazine is added, the NanoLuc emits light at 460 nm that can excite the Venus protein allowing for the detection of its emission at 525 nm by so-called bioluminescence resonance energy transfer (BRET).

The manuscript characterizes the sensor in vitro, in HeLa cells and in primary hippocampal neurons.

Fig. 1 illustrates the principles and validates the colocalization of the complemented Venus protein to ER and mitochondria. A challenge is that the NanoLuc luminescence-driven excitation of Venus is weak, but panel g-h demonstrate a dose dependency between the application of 10, 1 mM Ca^{2+} and EGTA. A further exploration of this should be performed to try to get an idea how low Ca^{2+} concentrations can be detected. This might be done by permeating the cells with Ca ionophores.

Fig. 2 demonstrates that the BRET signal predominantly is driven by the reduction in NanoLuc luminescence at 460 nm and not the Venus excitation at 525 nm. The signals can be activated by IP3 and histamine.

Fig. 3 demonstrates increased MAM formation due to co-expression of VAPB-PTPIP51 increases the BRET signal along with the Venus signal demonstrating a correlation between enhanced MAM formation and Ca²⁺ in the proximity of MAM. An attempt is made using staurosporine to increase Ca²⁺ flux through MAM. Staurosporine is a broad spectrum kinase inhibitor that induces apoptotic cell death. This will increase cytosolic Ca²⁺ in general and is not an optimal model for the purpose. The authors should make an effort to use another experimental paradigm where increased Ca²⁺ flux can be induced specifically through MAM.

Fig. 4 investigates the performance of the sensor in hippocampal mouse neurons. This analysis is important because it demonstrates that the BRET signal not simply correlates to the amount of MAM signal from Venus. The line scan in panels a and b support this claim but the data should be solidified by demonstrating the correlation of the signal to the neuronal processes. Essentially, the BRET signals appear as close to saturated signals, but it is not possible to localise them to the fluorescent EBFP2 signal that outlines the neuronal processes. An effort should be put into getting more precise information of the subcellular localization of the Venus signals and their associated BRET signals. This shall also be done for the histamine and Ca²⁺ induced signals in panels c-f.

Fig. 5-6 analyses the sensor in hippocampal neurons cultured from mouse models of Alzheimers disease (AD), ALS, PINK1 deficiency and human A53T-alpha-synuclein overexpression.

The analyses demonstrate the potential of the sensor by showing a correlated decrease in BRET and Venus signals in the AD model, selective decreased BRET signal in PINK-KO model, no effect in the ALS model and an increased BRET but decreased VENUS signal in the human A53T-alpha-synuclein model. This suggest the sensor holds potential for interrogating mechanisms in these diseases further.

Minor: The references to the mouse lines used should be given to the original publications and not by referencing colleagues.

Conclusively: the sensor holds promise but need a bit more analysis in the neurons to demonstrate is potential fully.

Reviewer #2 (Remarks to the Author):

The manuscript by Cho et al. describes the generation of a new BRET-based probe for measuring Ca²⁺ at mitochondria-associated ER membranes (MAMs) by applying the bimolecular fluorescence

complementation (BiFC) strategy. This sensor, termed MAM-Calflux, allows not only Ca²⁺ measurement but also to detect of quantitative structural changes that occur at MAM. This probe is unquestionably novel and exciting, and the paper is easy to follow. However, such BRET Ca²⁺ measurements it seems that cannot be calibrated, representing an essential pitfall for a Ca²⁺ indicator. I report my concerns here:

MAM Ca²⁺ measurements are not very informative without combining cytoplasmic, mitochondrial, and ER Ca²⁺ measurements. The lack of this information prevents me from reaching conclusions on the actual sensitivity and specificity of the developed probe. Also, the engineering of this probe could have significantly affected its KD for Ca²⁺. Authors should verify the KD value for this probe as this is essential information to understand its function.

- Figure 1: The strategies for investigating organelle contacts based on complementation strategies have the side effects of potentially generating artifactual connections. The overexpression of MAM-Calflux on the MAM formation should be tested by standard techniques (e.g., immunofluorescence colocalization analysis, PLA assay, and subcellular fractionation). Minor point: in Figure 1e, "3D remodeling" should be "3D rendering".

- Figure 2: Ip₃ is not cell-permeable, and since it seems that the experiment is run on non-permeabilized cells, the [Ca²⁺]_{mam} measured in this figure (and other) in response to stimulation of Ip₃ cannot be accounted for Ca²⁺ release from the ER. It is reported nonetheless that Ip₃ stimulation can elicit Ca²⁺ entry from the plasma membrane. This means that MAM-Calflux can also report variations in [Ca²⁺]_c. This should be clearly stated in the main text. In general, the [Ca²⁺]_{mam} is said to be way higher than [Ca²⁺]_c during an agonist stimulation (see PMID 9624056). Authors should confirm that MAM-Calflux reports more elevated [Ca²⁺]_{mam} compared to an equivalent probe located to the cytosol (e.g., native Calflux)

- Figure 3: The authors overexpress the ER-mitochondria tether VAPB and PTPIP51 to favor the formation of MAMs. The increased ER-mitochondria association has been revealed by MAM-Calflux. Augmented MAM integrity has also been observed with STS (by the way, the terms integrity does not seems meaningful, they would prefer "amount" and "extent").

These results are indeed all astonishing. First, I cannot understand why, at a steady state, the increase in MAM (amount or extension) should increase the [Ca²⁺]_{mam}. First, this result could be artifactual. If VAPB and PTPIP51 overexpression increases the number of contact sites, then this could lead to an increase in total BRET signal just because there is a higher chance that the probe couples. In other words, having a higher amount of acceptor available increases the BRET signal despite variations in [Ca²⁺]_{mam}. This means that MAM-Calflux is a better reporter of MAM amount than of [Ca²⁺]_{mam}. Authors should quantify the BRET variation in response to agonist stimulation (histamine NOT Ip₃) to

evaluate the impact of VAPB1 and PTPIP51 in $[Ca^{2+}]_{mam}$. Second, authors should exclude that this is not a variation in $[Ca^{2+}]_c$. Third, this may be an artifactual result due to the image quantitation technique. According to the materials and methods section, the BRET signal and Venus fluorescence signal are quantified by the use of ROIs on the image (it's unclear if the total intensity or average intensity was used) without any thresholding strategy. So if VAPB1 + PTPIP51 increases the area of the image where the BRET can occur, this could lead to an apparent increase in the BRET signal. Authors should mask their ideas to the Venus signal and report BRET as an average intensity within the Venus mask.

The increase in MAM amount by staurosporine is indeed astonishing. This result should be accompanied by quantitative data about the rise in MAM, performing immunofluorescence colocalization analysis, PLA assay, and subcellular fractionation.

Also, The authors analyzed the VAPB-PTPIP51 tethers, but other tethering factors should be tested to validate the actual efficacy of the probe.

- Figure 5: The data shown in the figure are not very informative since the changes in the MAM contact size are not related to potential changes in the mitochondrial network and the expression of ER-mitochondrial tethers. The authors should integrate the study, performed in the different pathological neurons, with ER and mitochondrial morphometric analysis, also controlling the expression of principal ER-mitochondria tethers.

- Figure 6D: Why does the kinetics reported in this figure dissimilar to figure 2B? Why does the $[Ca^{2+}]_{mam}$ kinetic not decline after IP3 stimulation?

This confirms that MAM-Caflux is also sensitive to intracellular Ca^{2+} -entry or Ca^{2+} variations in other intracellular districts.

- Finally, the BiFC complementation strategy would theoretically work with FRET-based Calcium reporters (e.g., the D3CPV series), which offers better performance, at least in live imaging experiments. Can the author explain why a BRET-based reporter is a better option?

Reviewer #3 (Remarks to the Author):

Cho et al. describe an elegant new sensor design to measure sub-cellularly defined calcium dynamics in cells. Here they split the Calflux sensor within the BRET acceptor fluorophore (Venus), allowing them to express the two halves of the sensor at different subcellular locations. They can then achieve proximity-dependent reconstitution of the sensor at mito-ER contact sites, enabling calcium measurements (and static labeling) within a specific subcellular compartment. They then use their new tool to measure calcium dynamics at mito-ER contact sites in response to various external stimuli (histamine/ionomycin), demonstrating that the tool is functional. They argue that the tool possesses dual functionality, in that direct excitation of Venus allows you to visualize static mito-ER contacts (due to Venus reconstitution), while the BRET ratio allows you to visualize calcium dynamics. They show that when inducing mito-ER contacts, this increases both the basal BRET ratio and the Venus excitation. They then show in a few different cell lines that the sensor can measure both calcium dynamics and mito-ER contact sites. Perhaps the most interesting biological finding they demonstrate is that in a PD model, they use their sensor to find that there is an excess of calcium at mito-ER contacts, perhaps due to an impaired ability of the mitochondria to uptake calcium. This is despite the fact that there are "looser" mito-ER contact sites in this cell line. They elegantly validate this finding by showing that a mito-targeted GCaMP6 demonstrates a reduction in calcium response in the PD cells, while an ER-targeted calcium indicator shows no changes in calcium response in the PD cells relative to wt cells.

Overall, the study demonstrates an appropriate and needed use case of their new sensor. The sensor design is innovative, and could allow new studies at any potential sub-compartment (particularly those that are not bound by a membrane). But given the emphasis of the study on the tool, rather than biology, I would have liked to see more thorough and rigorous characterization of the tool itself, as highlighted below.

Major points:

1. The authors acknowledge that split fluorescent proteins have known to spontaneously reconstitute, even in the absence of forced proximity. A concern would thus be that the MAM-Calflux increases mito-ER contacts in a non-physiological manner. While the authors do show that STS treatment/VAPB&PTPIP51 transfection can increase apparent MAM-Calflux (based on Venus excitation), there is no characterization as to how much the MAM-Calflux itself increases mito-ER contacts. It would be helpful if they could characterize this, perhaps by analyzing data as in Figure 1d co-staining TOM20/Calnexin juxtaposition in MAM-Calflux+ and MAM-Calflux- cells.

2. One concern for using the Venus excitation as a measure of mito-ER contacts is the wide variability in expression levels that the tool will have from cell to cell using transient transfection. While in theory the BRET responses should be independent of MAM-Calflux expression levels, the Venus excitation will not be. I would think that the variability in MAM-Calflux expression levels from cell to cell would outweigh the ~1.3-1.5 fold difference in Venus excitation observed in control versus experimental conditions in Figure 3. Another way potentially to address this concern would be in Figure 3g-l, to show in the same

cells vehicle treatment, followed by STS treatment, to demonstrate the relative increases in Venus expression as a result of this experimental treatment.

3. I did not find Figure 4 to be particularly convincing, as these were just a few example cells illustrated, and even then, the differences in BRET ratio do not really seem to be that localized nor large in magnitude (within 1.5-fold...). This might be better as a supplemental figure, and de-emphasized in the text.

Minor points:

1. It does not appear to be stated anywhere in the methods the amount of plasmid transfected in any of the conditions.
2. Why is the increase in Em525 so much smaller than the decrease in Em460 intensity in Figure 2d? Is this a similar observation with full-length Calflux? You almost don't even need the Em525 channel, other than as a background drift normalization.
3. Do you have a better example image for Figure 2a? In this example, it looks like the Em525 decreases immediately after IP3 treatment, which is not representative of panel e.

Reviewer #1 (Remarks to the Author):

Review of NatComm paper.

Inter-organellar contacts between ER and mitochondria (MAM) are considered of importance for a host of cellular functions. They govern the calcium (Ca²⁺) flux between ER and mitochondria, that is driven by the high Ca²⁺ concentration in ER and regulated by the IP3R channel in ER, the VDAC channel in mitochondria and the assembly complex between the organelles. The MAM are dynamic structures that can be assembled and disassembled at a fast pace, and their activity, e.i. the flux of Ca²⁺ difficult to measure.

The manuscript by Cho et al., describes the creation of a novel sensor based on the bimolecular complementation of a split fluorescent Venus protein with one half attached to a ER localization segment from Sac1 and the other half comprising a mitochondrial localization segment from AKAP, along with a Ca²⁺-binding domain from troponin C and a luminescent luciferase construct.

This assembly aims to tackle two questions. First, the localization of MAM to be visualized by fluorescence upon excitation of the complemented Venus protein, that can assemble when ER and mitochondrial membranes are in close proximity. Second, to measure the Ca²⁺ level in close proximity to the MAM, the binding of Ca²⁺ to the troponin C domain change its conformation whereby the NanoLuc domain is brought into proximity of the Venus protein. When furimazine is added, the NanoLuc emits light at 460 nm that can excite the Venus protein allowing for the detection of its emission at 525 nm by so-called bioluminescence resonance energy transfer (BRET).

The manuscript characterizes the sensor in vitro, in HeLa cells and in primary hippocampal neurons.

Fig. 1 illustrates the principles and validates the colocalization of the complemented Venus protein to ER and mitochondria. A challenge is that the NanoLuc luminescence-driven excitation of Venus is weak, but panel g-h demonstrate a dose dependency between the application of 10, 1 mM Ca²⁺ and EGTA. A further exploration of this should be performed to try to get an idea how low Ca²⁺ concentrations can be detected. This might be done by permeating the cells with Ca ionophores.

→ In response to the reviewer's comment, we moved the **previous Fig 1g-h** to **Supplementary Fig 2a-b** and added calcium dose-dependent wavelength scan results from in vitro purified proteins and live HeLa cells in **Fig 1g-h** and **Supplementary Fig 2c-d**, respectively.

→ In the **Previous Fig 1g-h** (now moved to **Supplementary Fig 2a-b**), the analysis was performed using live MAM-Calflux-expressing HeLa cells with 2 μM ionomycin and a designated dose of CaCl₂ or EGTA. To address the reviewer's concern regarding the weak BRET-mediated Venus signals, we increased the CaCl₂ concentration range (from 0.001 mM to 100.0 mM) and assessed under the same experimental conditions (**Supplementary Fig 2c-d**). Moreover, we prepared in vitro purified split-versions of CalfluxVTN to directly evaluate the Ca²⁺ response of the sensor at various concentrations of free-Ca²⁺ ions. The sensor exhibited a dramatic and clear dose-dependent BRET increase with approximately 209.4 nM for the estimated K_D value (**Fig 1g-h**). This result indicates that the split-form of Calflux (including MAM-Calflux) can distinguish differences between sufficiently low concentrations such as 17 and 38 nM. We also cautiously assume that the blunted Ca²⁺ response of MAM-Calflux in HeLa cells results from MAM Ca²⁺ concentration not being similar to the external/cytosolic Ca²⁺ concentration, even though MAM is not an entirely membrane-closed region. We revised the corresponding Results and Methods sections (*page 5, lines 118-121*,

and page 17, lines 407-428).

Fig 1g-h

Supplementary Fig 2c-d

→ Regarding the issue of cell permeabilization, the original data was acquired using ionomycin to permeabilize cells together with CaCl_2 treatment. We elaborated on figure panels and the corresponding Results section (*page 5, line 110*), as well as Figure legends clearly describing the use of ionomycin.

Fig. 2 demonstrates that the BRET signal predominantly is driven by the reduction in NanoLuc luminescence at 460 nm and not the Venus excitation at 525 nm. The signals can be activated by IP3 and histamine.

→ As it is natural that the BRET ratio is enhanced by the simultaneous decrease of luciferase signal (Em460) and increase of BRET-based signal (Em525), we agree that the *previous Fig 2b-e* showed an exaggerated Em460 decrease and an underrated Em525 increase. We suppose that some experimental conditions influenced the estimation of these signals. When acquiring microscopic images, we often experienced variations of signal intensities under various experimental conditions, including objective lenses, MAM-Cafluff expression levels, and substrate (furimazine) concentrations. These differences in signal intensities might cause different signal-to-noise ratios. In addition, when cell density and transfection rate were particularly high, we experienced a fast decrease of Em460 and Em525 signals, possibly due to the increased substrate exhaustion. The rapid decline of signals may have exaggerated the drop of Em460 signals and blunted the increase of Em525 signals. Thus, we put extra care into setting the experimental conditions to control sample-by-sample or set-by-set variations within a narrow range.

→ In response to the comment, we replaced *Fig 2a-g* with newly performed experimental data, demonstrating an apparent increase of Em525 and a decrease of Em460. In this experimental setting, we carefully controlled cell density, transfection rate, imaging buffers volume, and ROIs selection for this time-lapse imaging. Accordingly, we revised the Methods section corresponding to these experimental details (*page 16, lines 375-380 and 389-391, page 18, lines 436-438 and 440-444, and*

Fig 2a-g

Fig. 3 demonstrates increased MAM formation due to co-expression of VAPB-PTPIP51 increases the BRET signal along with the Venus signal demonstrating a correlation between enhanced MAM formation and Ca^{2+} in the proximity of MAM. An attempt is made using staurosporine to increase Ca^{2+} flux through MAM. Staurosporine is a broad spectrum kinase inhibitor that induces apoptotic cell death. This will increase cytosolic Ca^{2+} in general and is not an optimal model for the purpose. The authors should make an effort to use another experimental paradigm where increased Ca^{2+} flux can be induced specifically through MAM.

- To address the reviewer's point regarding staurosporine (STS), we moved STS treatment data to **Supplementary Fig 7** and revised the Results section (page 7, lines 167-168).
- To strengthen the relationship between MAM structural changes and MAM Ca^{2+} influx, we included experimental conditions with other tethering proteins such as GRP75 over-expression (**Fig 3e-h**) and MFN2 knockdown (**Fig 3i-l**). While the VAPB-PTPIP51 complex directly tethers mitochondrial and ER membranes, GRP75 joins with Ca^{2+} channels (IP3R and VDAC) and enhances MAM structure formation¹⁻⁴. In contrast, the knockdown of MFN2 has been one of the approaches to loosen MAM structure⁵⁻⁹. Altogether, three different experimental conditions modulating MAM tethers supported that MAM Ca^{2+} levels measured by MAM-Calflux largely correlate with the MAM structure closeness.

Fig 3e-l

Fig. 4 investigates the performance of the sensor in hippocampal mouse neurons. This analysis is important because it demonstrates that the BRET signal not simply correlates to the amount of MAM signal from Venus. The line scan in panels a and b support this claim but the data should be solidified by demonstrating the correlation of the signal to the neuronal processes. Essentially, the BRET signals appear as close to saturated signals, but it is not possible to localise them to the fluorescent EBFP2 signal that outlines the neuronal processes. An effort should be put into getting more precise information of the subcellular localization of the Venus signals and their associated BRET signals. This shall also be done for the histamine and Ca²⁺ induced signals in panels c-f.

→ In response to the reviewer's suggestion, we investigated MAM-Calflux signals from MAM puncta at the neuronal processes (axons and dendrites). By plotting data of 595 puncta from 27 individual neurons, we observed a positive correlation between the mean BRET ratios and the distances from the center of soma or the areas of MAM puncta, showing no correlation between BRET and Venus intensities (MAM closeness) (**Supplementary Fig 9a-c**). When we closely explored MAM puncta with strong BRET signals, distant from the soma, we noticed that many were located at the branching points of the neuronal processes (**Fig 4a-e**). Indeed, when we categorized MAM puncta by their locations (branching and non-branching), MAMs at branching points exhibited a significant correlation with distances from the soma (**Fig 4f-g**). Combined with previous reports on the importance of the local distribution of mitochondria and ER and local Ca²⁺ buffering in developing neurites¹⁰⁻¹³, our findings support the notion that local MAM Ca²⁺ can regulate the neurite structure formation. This also supports the potential of MAM-Calflux uncovering new biological roles of MAM Ca²⁺. This information was added in **Fig 4** and **Supplementary Fig 9**. The *previous Fig 4* was moved to **Supplementary Fig 8**, accordingly. We also revised the corresponding Results and Discussion sections (*page 8, lines 198-208, and pages 11-12, lines 283-288*).

→ Furthermore, we also measured the lysosome signal marked by mCherry-LAMP1 and MAM signals in parallel. Since LAMP1 signals had a very weak positive correlation with MAM Ca²⁺ levels, it was noticeable that BRET signals of MAM puncta that contain high LAMP1 signals showed a pattern converging to the mean BRET ratio in the soma (**Author response figure 1**). Notably, LAMP1-positive MAM puncta were located closer to soma and had a larger size than that of MAM alone, implying a potential role of LAMP1 in regulating ER-mitochondria contact and stabilizing Ca²⁺

levels. Although further in-depth studies are needed for this to be fully understood, these findings also support the potential of MAM-Calflux in understanding the intracellular distribution of MAM structure and MAM Ca^{2+} dynamics in various cellular processes.

→ Regarding the reviewer's concern about saturated BRET signals, we re-analyzed all the BRET quantitation data by masking the MAM structures based on Venus signals and measuring BRET signals only in masked MAM regions. We could have estimated more accurate MAM Ca^{2+} conditions under various conditions, including the steady-state intracellular distribution and IP_3 /histamine-mediated influx.

Fig 4

Supplementary Fig 9

Author response figure 1. The local distribution of MAM-specific calcium dynamics and lysosomal signals.

Comparison between MAM puncta overlapped with lysosome (LAMP1) signals and others. Bar graph (a) represents mean BRET ratio ($n = 27$ neurons) and scattered plot (b) represents correlation between normalized BRET signals and mean mCherry-LAMP1 signals, normalized to the soma. Bar graphs representing distance from center of soma (c), MAM area (d), and mean Venus excitation intensities (e). ($n = 27$ neurons) Scattered plot (f) representing correlation between BRET signals and distance to soma among the MAM puncta overlapping with lysosome (blue) and others (gray).

All results are presented as box plots representing median and interquartile range with whiskers min/max value and the cross representing mean value. All P-values were calculated using two-tailed Student's t-test for (a) and (c-e). For (b) and (f), each dot represents each MAM puncta located at the neurites ($n = 429$ and 166 for LAMP1-negative and LAMP1-positive, respectively, MAM puncta).

Fig. 5-6 analyses the sensor in hippocampal neurons cultured from mouse models of Alzheimers disease (AD), ALS, PINK1 deficiency and human A53T-alpha-synuclein overexpression.

The analyses demonstrate the potential of the sensor by showing a correlated decrease in BRET and Venus signals in the AD model, selective decreased BRET signal in PINK-KO model, no effect in the ALS model and an increased BRET but decreased VENUS signal in the human A53T-alpha-synuclein model. This suggest the sensor holds potential for interrogating mechanisms in these diseases further.

→ We appreciate reviewer's comment.

Minor: The references to the mouse lines used should be given to the original publications and not by referencing colleagues.

→ As suggested, we replaced all references to the mouse lines in the Methods section with those of original publications.

Conclusively: the sensor holds promise but need a bit more analysis in the neurons to demonstrate is potential fully.

Reviewer #2 (Remarks to the Author):

The manuscript by Cho et al. describes the generation of a new BRET-based probe for measuring Ca²⁺ at mitochondria-associated ER membranes (MAMs) by applying the bimolecular fluorescence complementation (BiFC) strategy. This sensor, termed MAM-Calflux, allows not only Ca²⁺ measurement but also to detect of quantitative structural changes that occur at MAM. This probe is unquestionably novel and exciting, and the paper is easy to follow. However, such BRET Ca²⁺ measurements it seems that cannot be calibrated, representing an essential pitfall for a Ca²⁺ indicator. I report my concerns here:

MAM Ca²⁺ measurements are not very informative without combining cytoplasmic, mitochondrial, and ER Ca²⁺ measurements. The lack of this information prevents me from reaching conclusions on the actual sensitivity and specificity of the developed probe. Also, the engineering of this probe could have significantly affected its K_D for Ca²⁺. Authors should verify the K_D value for this probe as this is essential information to understand its function.

→ In response to the reviewer's comment, we prepared in vitro purified split-versions of CalfluxVTN in order to directly evaluate the Ca²⁺ response of the sensor under various concentrations of free-Ca²⁺ ions (from 0.017 to 39.00 μM). The sensor exhibited a significant dose-dependent BRET increase with approximately 209.4 nM for the estimated K_D value and 1.049 for the hill coefficient (**Fig 1g-h**). Notably, the original (non-split version) CalfluxVTN exhibited similar kinetics (K_D value 480 nM and Hill coefficient 1.36)¹⁴. We avoided direct quantitative comparison of the datasets from two different versions of Calflux due to the completely different experimental conditions used. We moved *previous Fig 1g-h* to *Supplementary Fig 2a-b* and added calcium dose-dependent wavelength scan results from in vitro purified proteins in **Fig 1g-h**. Accordingly, we revised the corresponding Results and Methods sections (*page 5, lines 118-121, and page 17, lines 407-424*).

Fig 1g-h

- Figure 1: The strategies for investigating organelle contacts based on complementation strategies have the side effects of potentially generating artifactual connections. The overexpression of MAM-Calflux on the MAM formation should be tested by standard techniques (e.g., immunofluorescence colocalization analysis, PLA assay, and subcellular fractionation).

→ In response to the reviewers' concern about the effect of MAM-Calflux on the MAM structure, we added ER-mitochondria colocalization analysis data in *Supplementary Fig 1e*, which validated that short-term (17 h) expression of MAM-Calflux has a minimal effect on MAM structures. However, importantly, its long-term (more than 48 h) expression certainly enhanced colocalization, which confirmed the previous reports on the intrinsic self-assembly issue of the BiFC strategy.

Supplementary Fig 1e

→ As we mentioned earlier in the Discussion section, a long-term over-expression of MAM-Calflux caused enhanced MAM structure signals, high colocalization coefficients, and occasional aggregation-like signals in some cells. This phenomenon usually initiates 24-36 h after transfection. Therefore, we optimized experimental conditions, in particular, by controlling cell density, strictly restricting assay time after transfection, and using relatively small amounts of plasmid DNAs. It is noteworthy that a decreased MAM structure from MFN2 knockdown cells was effectively detected upon the MAM-Calflux expression, indicating that the MAM-Calflux self-assembly effect is not significantly compromising the results in experimental conditions used in our analyses. Nonetheless, further technical improvements to alleviate the potential issues of self-assembly can further enhance the usefulness of the MAM-Calflux.

→ Accordingly, we revised the corresponding Results, Discussion, and Methods sections to present our validation results, discuss this issue, and provide the optimized experimental conditions (*pages 4-5, lines 103-107, page 13, lines 332-334 and 337-338, page 16, lines 375-380 and 389-391, and page 18, lines 436-444*).

Minor point: in Figure 1e, "3D remodeling" should be "3D rendering".

→ We thank the reviewer for pointing this out. We revised the text in **Fig 1e**.

- Figure 2: Ip_3 is not cell-permeable, and since it seems that the experiment is run on non-permeabilized cells, the $[Ca^{2+}]_{mam}$ measured in this figure (and other) in response to stimulation of Ip_3 cannot be accounted for Ca^{2+} release from the ER. It is reported nonetheless that Ip_3 stimulation can elicit Ca^{2+} entry from the plasma membrane. This means that MAM-Calflux can also report variations in $[Ca^{2+}]_c$. This should be clearly stated in the main text. In general, the $[Ca^{2+}]_{mam}$ is said to be way higher than $[Ca^{2+}]_c$ during an agonist stimulation (see PMID 9624056). Authors should confirm that MAM-Calflux reports more elevated $[Ca^{2+}]_{mam}$ compared to an equivalent probe located to the cytosol (e.g., native Calflux)

→ We apologize for the unclear description of the experimental conditions. We performed all experiments with IP_3 or $CaCl_2$ treatment with ionomycin final concentration 2-20 μM . We revised figure panels and texts, including the Results (*page 5, lines 110-111, and page 6, line 140*), Methods (*page 18, lines 427-428, and page 19, lines 466-468*), and Figure legends to describe the use of ionomycin clearly.

→ In response to the reviewer's suggestion regarding the equivalent probe located in the cytosol, we generated the split-version of CalfluxVTN (cyto-spCalflux) lacking ER/mitochondria targeting and linker sequences (**Supplementary Fig 6a**). When treated with histamine to induce ER Ca²⁺ release, MAM-Calflux exhibited an enhanced Ca²⁺ response upon the VAPB-PTPIP51 expression (**Supplementary Fig 5e-f**), whereas cyto-spCalflux did not (**Supplementary Fig 6e-f**). This further supports the notion that the MAM-Calflux response is relatively specific to the change in MAM Ca²⁺ level.

Supplementary Fig 5e-f

Supplementary Fig 6a,e-f

- Figure 3: The authors overexpress the ER-mitochondria tether VAPB and PTPIP51 to favor the formation of MAMs. The increased ER-mitochondria association has been revealed by MAM-Calflux. Augmented MAM integrity has also been observed with STS (by the way, the terms integrity does not seems meaningful, they would prefer "amount" and "extent").

→ When we quantitated MAM structures, we used two variables: the area of the MAM marker (MAM area) and fluorescence intensity (probably reflecting the closeness or compactness of MAM). To address the reviewer's suggestion, we replaced the phrase 'MAM integrity' with 'MAM extent' or 'MAM closeness.' (page 7, lines 160 and 166, page 8, line 186, page 9, line 219, and page 10, line 247)

These results are indeed all astonishing. First, I cannot understand why, at a steady state, the increase in MAM (amount or extension) should increase the [Ca²⁺]_{mam}. First, this result could be artifactual. If VAPB and PTPIP51 overexpression increases the number of contact sites, then this could lead to an increase in total BRET signal just because there is a higher chance that the probe couples. In other words, having a higher amount of acceptor available increases the BRET signal despite variations in [Ca²⁺]_{mam}. This means that MAM-Calflux is a better reporter of MAM amount than of [Ca²⁺]_{mam}.

Authors should quantify the BRET variation in response to agonist stimulation (histamine NOT Ip3) to evaluate the impact of VAPB1 and PTPIP51 in $[Ca^{2+}]_{mam}$. Second, authors should exclude that this is not a variation in $[Ca^{2+}]_c$.

- We hypothesized that the steady-state MAM Ca^{2+} levels would be determined by multiple factors, including the amount and frequency of ER Ca^{2+} release, the amount of mitochondrial Ca^{2+} uptake, and the amount of Ca^{2+} diffused into the cytosol. Although addressing all these potential factors is beyond this study's scope, we partly explored this point by targeting MAM tethering proteins to modulate MAM structure as MAM structure by enhanced tetherings affects either ER Ca^{2+} release or mitochondrial Ca^{2+} uptake or both^{9,15,16}. The tightened/loosened MAM structure is often accompanied by increased/reduced ER Ca^{2+} release, supporting that MAM tethering proteins can affect Ca^{2+} flux in MAM by stimulating ER Ca^{2+} release. Notably, *Zampese et al.* reported that ectopic Presenilin 2 expression enhanced both ER-mitochondria colocalizations (MAM structure) and Ca^{2+} hotspots near mitochondria (supposedly MAM Ca^{2+}), utilizing their OMM-specific Ca^{2+} sensor, N33D1cpv¹⁷. This also supports the relationship between MAM structure closeness and MAM Ca^{2+} levels. Indeed, we observed that steady-state MAM Ca^{2+} levels were increased along with the upregulated MAM structure by VAPB-PTPIP51 tethers or the apoptotic stimulus such as STS (**Fig 3a-d** and **Supplementary Fig 7**). To address reviewer's comments, we also tested GRP75 over-expression (increasing IP3R-VDAC interaction, **Fig 3e-h**) and MFN2 knockdown (decreasing mitochondria-ER contacts and mitochondrial fusion, **Fig 3i-l**).
- Regarding the reviewer's concern about "a higher amount of acceptor available increases the BRET signal," we agree that it could affect BRET signals. However, we would like to emphasize that MAM-Calflux responds to MAM Ca^{2+} changes upon IP₃/histamine/ $CaCl_2$ -induced Ca^{2+} influx without an increase in MAM structure (**Fig 2** and **Supplementary Fig 5**). Furthermore, an enhanced BRET signal was observed even in SNCA*A53T tg neurons with decreased MAM structure. Moreover, when we examined some concentrated MAM signals in neurites, there was little correlation between BRET and Venus signals (Pearson's coefficient $r = 0.0809$, **Supplementary Fig 9c**). This also supports the notion that MAM-Calflux is effectively measuring MAM Ca^{2+} dynamics. In response to the reviewer's comments, we have discussed the reviewer's point in the Discussion section (*page 13, lines 332-334*).
- In response to the reviewer's suggestion, we also assessed histamine-induced Ca^{2+} changes in MAM, ER, mitochondria, and the cytosol upon VAPB-PTPIP51 expression (**Supplementary Fig 5e-l** and **Supplementary Fig 6e-f**). While mitochondrial Ca^{2+} influx was increased and MAM-Calflux detected increased Ca^{2+} influx by tether expression, there was no significant change in cytosolic Ca^{2+} influx and ER Ca^{2+} release. This indicates that MAM-Calflux detects MAM Ca^{2+} levels.
- To address the reviewer's comment on the possibility of a variation in $[Ca^{2+}]_c$, we measured cytosolic Ca^{2+} levels through the cyto-spCalflux, a cytosolic version of MAM-Calflux (**Supplementary Fig 6a-c**). Indeed, cyto-spCalflux exhibited little changes in BRET signals upon VAPB-PTPIP51 expression, indicating a minimal effect of cytosolic Ca^{2+} in this condition. This also supports the notion that enhanced MAM-Calflux BRET signal upon tether protein expression reflect enhanced steady-state MAM Ca^{2+} levels.

Supplementary Fig 5e-l

Supplementary Fig 6

Third, this may be an artifactual result due to the image quantitation technique. According to the materials and methods section, the BRET signal and Venus fluorescence signal are quantified by the use of ROIs on the image (it's unclear if the total intensity or average intensity was used) without any thresholding strategy. So if VAPB1 + PTPIP51 increases the area of the image where the BRET can occur, this could lead to an apparent increase in the BRET signal. Authors should mask their ideas to the Venus signal and report BRET as an average intensity within the Venus mask.

→ In response to the reviewer's suggestion, we re-analyzed all the BRET quantitation data by masking the MAM (Venus) signals and measuring BRET signals only in masked MAM regions. An example of the image processing steps is presented in **Supplementary Fig 3b-c**. As a result, we could define a mean BRET ratio reflecting MAM Ca^{2+} concentration. Additionally, in this re-analysis, we quantitated Venus excitation signals using two parameters, MAM area (area of Venus) and MAM extent (total intensity of Venus), for a better presentation of the data. Regarding this, we revised text

materials, including the Results (page 5, lines 125-127) and Methods (page 19, lines 456-458).

Supplementary Fig 3b-c

→ In the re-analysis (*Fig 3a-d*), we observed that VAPB-PTPIP51 expression increased MAM Ca^{2+} levels, MAM area, and MAM extent, similar to data in *previous Fig 3a-c*. Similarly, re-analysis of the other previous data did not change the tendencies of the data from STS treatment (*previous Fig 3g-l*, now *Supplementary Fig 7*), 5xFAD neurons (*previous Fig 5a-c*, now *Fig 5a-d*), and SNCA*A53T tg neurons (*previous Fig 6a-e*, now *Fig 6a-f*). Notably, GRP75, another tethering protein, increased both MAM Ca^{2+} levels and MAM extent but did not significantly change the MAM area (*Fig 3e-h*). We hypothesize that this difference between VAPB-PTPIP51 and GRP75 arose from their functional difference in MAM; VAPB-PTPIP51 are membrane-anchored proteins that directly tether ER-mitochondria membranes¹⁸⁻²⁰, whereas GRP75 binds to IP3R and VDAC1 to maintain/enhance MAM structure indirectly¹⁻⁴. Therefore, MAM-Calflux can discriminate changes in MAM Ca^{2+} levels, MAM area, and MAM extent, as also seen in SNCA*A53T tg neurons.

Fig 3a-h

Previous Fig 3a-c

The increase in MAM amount by staurosporine is indeed astonishing. This result should be accompanied by quantitative data about the rise in MAM, performing immunofluorescence colocalization analysis, PLA assay, and subcellular fractionation. Also, The authors analyzed the VAPB-PTPIP51 tethers, but other tethering factors should be tested to validate the actual efficacy of the probe.

→ We acknowledge the reviewers' suggestion regarding STS, and we have moved STS treatment results to **Supplementary Fig 7**, and revised the Results section (page 7, lines 167-168). Indeed, short-term (1-2 h) treatment of STS induces early phase apoptosis along with increased intracellular Ca^{2+} influx²¹, and it was already directly and indirectly verified to increase MAM formation²²⁻²⁶. We also confirmed the increased ER-mitochondria contact by STS treatment using colocalization assay and time-lapse imaging of MAM-Calflux without furimazine (**Author response figure 2**).

Author response figure 2. Increased ER-mitochondria colocalization and Venus excitation intensities of MAM-Calflux by STS treatment.

(a) Representative images (left) and Manders' overlap coefficients (right) of colocalization between mitochondria (TOM20) and ER (Calreticulin) among HeLa cells treated with 1 μ M STS for 2 hours. (n = 43 for control cells, n = 35 for STS treated cells)
 (b-c) Time-dependent relative intensities (a) of Venus from MAM-Calflux and EBFP2-N1 control in HeLa cells treated with 1 μ M STS, and peak amplitude at 120 min after STS treatment (b). (n = 164 cells)

The scale bar represents 10 μ m. All results are presented as box plots representing median and interquartile range with whiskers min/max value for (a) and (c), and as means \pm SEM for (b). The P-value was calculated using two-tailed Student's t-tests for (a) and (c).

→ To strengthen the relationship between MAM structural changes and MAM Ca^{2+} flux, we added data with other tethering proteins, such as GRP75 over-expression and MFN2 knockdown. VAPB-PTPIP51 complex directly tethers mitochondrial and ER membranes and results in an increase of both MAM formation and its Ca^{2+} level (**Fig 3a-d**). GRP75 joins with Ca^{2+} channels, IP3R and VDAC, indirectly enhancing MAM structure formation¹⁻⁴. MAM-Calflux evaluated that the MAM area was not changed, but the MAM extent and the Ca^{2+} level were enhanced by GRP75 expression (**Fig 3e-h**). In contrast, the knockdown of MFN2 has been one of the approaches to loosen MAM structure⁵⁻⁹. MAM-Calflux identified a reduction of both MAM formation and MAM Ca^{2+} levels by the MFN2 deficiency (**Fig 3i-l**). Altogether, three different experimental conditions modulating MAM tethers supported that MAM Ca^{2+} levels are largely correlated with the MAM structure closeness.

Fig 3

- Figure 5: The data shown in the figure are not very informative since the changes in the MAM contact size are not related to potential changes in the mitochondrial network and the expression of ER-mitochondrial tethers. The authors should integrate the study, performed in the different pathological neurons, with ER and mitochondrial morphometric analysis, also controlling the expression of principal ER-mitochondria tethers.

→ In response to the reviewer's suggestion, we introduced the VAPB-PTPIP51 tethering complex in neurodegenerative disease model neurons with reduced MAM structures: 5xFAD and SNCA*A53T. As expected, the colocalization assay showed that VAPB-PTPIP51 expression could restore the ER-mitochondria contacts in both mouse lines (**Fig 5e** and **Supplementary Fig 11a**). Consistently, MAM-Calflux indicated a restored MAM formation in this condition (**Fig 5f-i** and **Supplementary Fig 11b-e**). In contrast, reduced BRET signals in 5xFAD neurons were restored by expressing tethers, but the exaggeration of signals in SNCA*A53T neurons was maintained. This suggests that reduced basal MAM Ca^{2+} levels in 5xFAD were derived from loosened MAM structure. On the other hand, the increased MAM Ca^{2+} levels in SNCA*A53T were possibly determined by functional components

of MAM that determine the Ca^{2+} level in MAM relatively independent from MAM structure. We revised the corresponding Results and Discussion sections (page 9, lines 220-222, page 10, lines 238-239, and page 13, lines 306-307).

Fig 5e-i

Supplementary Fig 11

→ To further address the reviewer's point, we performed mitochondrial morphometric analysis from TOM20 immunostained neurodegenerative disease model neurons with or without VAPB-PTPIP51 tethers. Although it is preliminary, we found that the mitochondrial contents were reduced in both 5xFAD and SNCA*A53T neurons, and they were restored by the expression of tethers (*Author response figure 3*), as previously reported²⁷⁻²⁹. On the contrary, we could not detect a significant

difference in mitochondrial circularity and size. It is necessary to design a more sophisticated experimental setting to further investigate this issue, for instance, super-resolution microscopy or EM imaging analysis, which can more precisely determine the morphological changes of the organelles by evaluating the patterns and quantities of their contacts in disease model neurons.

Author response figure 3. Mitochondrial morphometric analysis of 5xFAD tg and SNCA*A53T tg mice neurons.

(a-c) Mitochondrial morphometric analysis of 5xFAD tg neurons with VAPB-PTPIP51 expression. (a) Mitochondrial content as percentage of mitochondrial (TOM20) area among total cell area. Mitochondrial morphology as circularity (b) and length (c) of mitochondrial puncta. (n = 51 for wild-type neurons, n = 43 for 5xFAD tg neurons, and n = 47 for VAPB-PTPIP51 expressing 5xFAD tg neurons)

(d-f) Mitochondrial morphometric analysis of SNCA*A53T tg neurons with VAPB-PTPIP51 expression. (d) Mitochondrial content as percentage of mitochondrial (TOM20) area among total cell area. Mitochondrial morphology as circularity (e) and length (f) of mitochondrial puncta. (n = 31 for wild-type neurons, n = 36 for 5xFAD tg neurons, and n = 39 for VAPB-PTPIP51 expressing 5xFAD tg neurons)

All results are presented as box plots representing median and interquartile range with whiskers min/max value. All P-values were calculated using one-way ANOVA with Bonferroni's multiple comparison test.

- Figure 6D: Why does the kinetics reported in this figure dissimilar to figure 2B? Why does the [Ca²⁺]_{mam} kinetic not decline after IP₃ stimulation?

This confirms that MAM-Caflux is also sensitive to intracellular Ca²⁺-entry or Ca²⁺ variations in other intracellular districts.

→ At first, we presented *previous Fig 2a-g* in time from -97.5 sec to 397.5 sec with IP₃ treatment at 0 sec; and *previous Fig 6d* (now *Fig 6e*) was between -22.5 sec and 67.5 sec after IP₃ treatment, to make it more consistent with mitochondrial (mito-GCaMP6s) Ca²⁺ influx and ER (R-CEPIA1-er) Ca²⁺ efflux graphs in *previous Fig 6f-i* (now *Fig 6g-j*). Indeed, the full time-lapse imaging graph (before 67.5 sec – after 292.5 sec) exhibits gradual recovery of MAM Ca²⁺ levels (*Author response figure 4*).

Author response figure 4. The full time-lapse imaging results in Fig 6e.

Exaggerated MAM Ca^{2+} influx was indicated by MAM-Calflux in response to 80 μM IP_3 in SNCA*A53T tg mouse neurons at DIV7. All results are presented as means \pm SEM. (n = 44 for wild-type neurons and n = 32 for SNCA*A53T tg neurons)

→ We would like to note that the experimental conditions for *previous Fig 6d (Fig 6e)* and *Fig 2a-g* were different. In the *previous Fig 2a-g*, the experiment was performed in HeLa cells and with 20 μM ionomycin and 5 μM IP_3 . In contrast, in the *previous Fig 6d (Fig 6e)*, the experiment was performed in primary cultured DIV7 hippocampal neurons with 20 μM ionomycin and 80 μM IP_3 . In our hands, neurons require higher concentrations of IP_3 than HeLa cells for detection of ER-mitochondria Ca^{2+} transfer and display prolonged Ca^{2+} flux.

→ In response to reviewers' comment, we replaced *Fig 2a-g* with newly performed experimental data with both low (10 μM) and high (80 μM) IP_3 concentrations, and we presented an apparent increase of Em525 and decrease of Em460 and prolonged BRET signals. In this condition, Ca^{2+} in HeLa still seem to decline faster than in neurons.

Fig 2a-g

→ In our experience, different cell lines show different basal MAM Ca^{2+} levels. For instance, HeLa cells exhibited approximately 0.4413 ± 0.0300 (*Fig 3b*) and 0.4571 ± 0.0190 (*Supplementary Fig*

7b), and neurons exhibited approximately 0.5074 ± 0.0379 (*Supplementary Fig 5b*) and 0.6855 ± 0.0695 (*Supplementary Fig 7f*), supposedly reflecting different Ca^{2+} physiology in these cell types. We suppose that some experimental conditions influenced the estimation of these signals. When acquiring microscopic images, we often experienced variations of signal intensities under various experimental conditions, including objective lenses, MAM-Calflux expression levels, and substrate (furimazine) concentrations. These differences in signal intensities might cause different signal-to-noise ratios, although images were processed by subtracting background signals before quantitation, and affected the ratio of changes by IP_3 or histamine treatment. In addition, when cell density and transfection rate were particularly high, we experienced a fast decrease of Em460 and Em525 signals, possibly due to the increased substrate exhaustion. The rapid decline of signals may have exaggerated the drop of Em460 signals and blunted the increase of Em525. Thus, we put extra-care to set the experimental conditions to control sample-by-sample or set-by-set variations within narrow range. For example, we tried to set the experimental conditions such as cell density, transfection conditions, MAM-Calflux expression levels, and imaging time from furimazine treatment, etc.

- Finally, the BiFC complementation strategy would theoretically work with FRET-based Calcium reporters (e.g., the D3CPV series), which offers better performance, at least in live imaging experiments. Can the author explain why a BRET-based reporter is a better option?

→ Both FRET- and BRET-based GECIs share the important advantage of ratiometric sensors. Meanwhile, BRET-GECIs have some advantages over FRET-GECIs because they use bioluminescence as the donor. First, BRET-GECIs do not require excitation illumination and thus have fewer photobleaching or autofluorescence background issues. Second, BRET-GECIs can be used with photo-active or photo-responsive molecules and tissues, such as in combination with optogenetic techniques or experimenting in the retina and chloroplast. Third, BRET-based biosensors provide more room for utilizing additional fluorescent molecules with a similar wavelength range. We were encouraged by a number of brilliant works by the Carl H Johnson group enjoying these advantages^{14,30,31}. We do agree that FRET-GECIs have advantages over BRET-GECIs especially in terms of signal intensities outperforming in the time-lapse live imaging, and will provide additional routes to look into Ca^{2+} dynamics in the MAM once it becomes available.

Reviewer #3 (Remarks to the Author):

Cho et al. describe an elegant new sensor design to measure sub-cellularly defined calcium dynamics in cells. Here they split the Calflux sensor within the BRET acceptor fluorophore (Venus), allowing them to express the two halves of the sensor at different subcellular locations. They can then achieve proximity-dependent reconstitution of the sensor at mito-ER contact sites, enabling calcium measurements (and static labeling) within a specific subcellular compartment. They then use their new tool to measure calcium dynamics at mito-ER contact sites in response to various external stimuli (histamine/ionomycin), demonstrating that the tool is functional. They argue that the tool possesses dual functionality, in that direct excitation of Venus allows you to visualize static mito-ER contacts (due to Venus reconstitution), while the BRET ratio allows you to visualize calcium dynamics. They show that when inducing mito-ER contacts, this increases both the basal BRET ratio and the Venus excitation. They then show in a few different cell lines that the sensor can measure both calcium dynamics and mito-ER contact sites. Perhaps the most interesting biological finding they demonstrate is that in a PD model, they use their sensor to find that there is an excess of calcium at mito-ER contacts, perhaps due to an impaired ability of the mitochondria to uptake calcium. This is despite the fact that there are "looser" mito-ER contact sites in this cell line. They elegantly validate this finding by showing that a mito-targeted GCaMP6 demonstrates a reduction in calcium response in the PD cells, while an ER-targeted calcium indicator shows no changes in calcium response in the PD cells relative to wt cells.

Overall, the study demonstrates an appropriate and needed use case of their new sensor. The sensor design is innovative, and could allow new studies at any potential sub-compartment (particularly those that are not bound by a membrane). But given the emphasis of the study on the tool, rather than biology, I would have liked to see more thorough and rigorous characterization of the tool itself, as highlighted below.

Major points:

1. The authors acknowledge that split fluorescent proteins have known to spontaneously reconstitute, even in the absence of forced proximity. A concern would thus be that the MAM-Calflux increases mito-ER contacts in a non-physiological manner. While the authors do show that STS treatment/VAPB&PTPIP51 transfection can increase apparent MAM-Calflux (based on Venus excitation), there is no characterization as to how much the MAM-Calflux itself increases mito-ER contacts. It would be helpful if they could characterize this, perhaps by analyzing data as in Figure 1d co-staining TOM20/Calnexin juxtaposition in MAM-Calflux+ and MAM-Calflux- cells.

→ To address the concern about the effect of MAM-Calflux on the MAM structure, we added ER-mitochondria colocalization analysis data in **Supplementary Fig 1e**, validating that short-term (17 h) expression of MAM-Calflux has a minimal effect on MAM structures. However, it is noteworthy that long-term (longer than 48 h) expression of MAM-Calflux did enhance colocalization, which is consistent with previous observations reflecting the intrinsic self-assembly issue of the BiFC strategy. Therefore, we optimized all our experimental conditions, in particular, by controlling cell density, strictly restricting assay time after transfection, and using a relatively small amount of plasmid DNAs for transfection.

Supplementary Fig 1e

- To enhance the instrumental value of MAM-Calflux, we suggest multiple approaches, although potential sensitivity issues may need to be taken care of alongside. First, the Venus fragment domains (VN173 and VC155 in MAM-Calflux) can be modified to those with fewer signal-to-noise ratios, such as VN210/VC210 combinations³², introducing point mutations (V150L, I152L, L201V, L207V)^{33,34}, or dimerization-dependent fluorescent proteins (ddGFP-AB, ddYFP-AB, ddRFP)³⁵. Second, low-expression promoters, such as UBC, can be used instead of the CMV promoter to minimize the artifactual assembly. Third, an inducible gene expression system, such as Tet-on or cumate-inducible systems, can be an option to control the MAM-Calflux expression.
- Reflecting these notions, we revised the corresponding Results, Discussion, and Methods sections to present our validation results, discuss this issue, and provide the optimized experimental conditions used in this study (pages 4-5, lines 103-107, page 13, lines 332-334 and 337-338, page 16, lines 375-380 and 389-391, and page 18, lines 436-444).

2. One concern for using the Venus excitation as a measure of mito-ER contacts is the wide variability in expression levels that the tool will have from cell to cell using transient transfection. While in theory the BRET responses should be independent of MAM-Calflux expression levels, the Venus excitation will not be. I would think that the variability in MAM-Calflux expression levels from cell to cell would outweigh the ~1.3-1.5 fold difference in Venus excitation observed in control versus experimental conditions in Figure 3. Another way potentially to address this concern would be in Figure 3g-l, to show in the same cells vehicle treatment, followed by STS treatment, to demonstrate the relative increases in Venus expression as a result of this experimental treatment.

- We agree that the MAM structure signals measured by Venus in MAM-Calflux are not suitable for the direct comparison of cell-to-cell differences, unlike BRET signals. Although limited, we still suggest applications of MAM-Calflux as a MAM marker. First, it can be used to analyze intracellular region-to-region differences from the single cell, as demonstrated in **Fig 4d-e** and **Supplementary Fig 9b-c**. Second, it is possible to investigate the changes in MAM structure in a time-dependent manner. Third, it is still applicable to the analysis of group-by-group differences with a sufficient number of cells, such as in **Fig 3**, **Fig 5**, and **Fig 6**.
- Regarding the reviewer's suggestion about demonstrating live signal increment of BRET and Venus by STS treatment, this was unfortunately unachievable by technical limitations; when cells treated with furimazine substrate, BRET signals were significantly decreased after approximately 30-40 min, likely due to relatively short half-life of furimazine (~105 min in HBSS+HEPES buffer without any enzyme³⁶, even faster with MAM-Calflux expressing cells). Because STS-induced MAM structure

formation took longer than this time range (about 1-2 h according to references and our experiences), it was technically challenging to set an experimental condition to analyze simultaneously both MAM Ca^{2+} and MAM structure in STS-treated cells. On the other hand, it was possible to measure the STS-induced changes of MAM extent via Venus excitations, which supports the potential of MAM-Calflux to detect relative changes in MAM structure (*Author response figure 2b-c*).

Author response figure 2. Increased ER-mitochondria colocalization and Venus excitation intensities of MAM-Calflux by STS treatment.

(a) Representative images (left) and Manders' overlap coefficients (right) of colocalization between mitochondria (TOM20) and ER (Calreticulin) among HeLa cells treated with 1 μM STS for 2 hours. (n = 43 for control cells, n = 35 for STS treated cells) (b-c) Time-dependent relative intensities (a) of Venus from MAM-Calflux and EBFP2-N1 control in HeLa cells treated with 1 μM STS, and peak amplitude at 120 min after STS treatment (b). (n = 164 cells)

The scale bar represents 10 μm . All results are presented as box plots representing median and interquartile range with whiskers min/max value for (a) and (c), and as means \pm SEM for (b). The P-value was calculated using two-tailed Student's t-tests for (a) and (c).

3. I did not find Figure 4 to be particularly convincing, as these were just a few example cells illustrated, and even then, the differences in BRET ratio do not really seem to be that localized nor large in magnitude (within 1.5-fold...). This might be better as a supplemental figure, and de-emphasized in the text.

→ In response to the reviewer's comment, we investigated MAM-Calflux signals from MAM puncta along the neuronal processes (axons and dendrites). By plotting data of 595 puncta from 27 individual neurons, we observed a positive correlation between the mean BRET ratios and the distances from the center of soma or the areas of MAM puncta, showing no correlation between BRET and Venus intensities (MAM closeness) (*Supplementary Fig 9a-c*). When we closely explored MAM puncta with strong BRET signals, distant from the soma, we noticed that many were located at the branching points of the neuronal processes (*Fig 4a-e*). Indeed, when we categorized MAM puncta by their locations (branching and non-branching), MAMs at branching points exhibited a significant correlation with distances from the soma (*Fig 4f-g*). Combined with previous reports on the importance of the local distribution of mitochondria and ER and local Ca^{2+} buffering in developing

neurites¹⁰⁻¹³, our findings support the notion that local MAM Ca^{2+} can regulate the neurite structure formation. This also supports the potential of MAM-Calflux uncovering new biological roles of MAM Ca^{2+} . This information was added in **Fig 4** and **Supplementary Fig 9**. The *previous Fig 4* was moved to **Supplementary Fig 8**, accordingly. We also revised the corresponding Results and Discussion sections (*page 8, lines 198-208, and pages 11-12, lines 283-288*).

→ Furthermore, we also measured the lysosome signal marked by mCherry-LAMP1 and MAM signals in parallel. Since LAMP1 signals had a very weak positive correlation with MAM Ca^{2+} levels, it was noticeable that BRET signals of MAM puncta that contain high LAMP1 signals showed a pattern converging to the mean BRET ratio in the soma (**Author response figure 1**). Notably, LAMP1-positive MAM puncta were located closer to soma and had a larger size than that of MAM alone, implying a potential role of LAMP1 in regulating ER-mitochondria contact and stabilizing Ca^{2+} levels. Although further in-depth studies are needed for this to be fully understood.

Fig 4

Supplementary Fig 9

Author response figure 1. The local distribution of MAM-specific calcium dynamics and lysosomal signals.

Comparison between MAM puncta overlapped with lysosome (LAMP1) signals and others. Bar graph (a) represents mean BRET ratio ($n = 27$ neurons) and scattered plot (b) represents correlation between normalized BRET signals and mean mCherry-LAMP1 signals, normalized to the soma. Bar graphs representing distance from center of soma (c), MAM area (d), and mean Venus excitation intensities (e). ($n = 27$ neurons) Scattered plot (f) representing correlation between BRET signals and distance to soma among the MAM puncta overlapping with lysosome (blue) and others (gray).

All results are presented as box plots representing median and interquartile range with whiskers min/max value and the cross representing mean value. All P-values were calculated using two-tailed Student's t-test for (a) and (c-e). For (b) and (f), each dot represents each MAM puncta located at the neurites ($n = 429$ and 166 for LAMP1-negative and LAMP1-positive, respectively, MAM puncta).

Minor points:

1. It does not appear to be stated anywhere in the methods the amount of plasmid transfected in any of the conditions.

→ We apologize for not including this in the original manuscript, and we have revised the corresponding Methods section (*page 16, lines 375-380 and 389-391*) with more details.

2. Why is the increase in Em525 so much smaller than the decrease in Em460 intensity in Figure 2d? Is this a similar observation with full-length Calflux? You almost don't even need the Em525 channel, other than as a background drift normalization.

→ As it is natural that the BRET ratio is enhanced by the simultaneous decrease of luciferase signal (Em460) and increase of BRET-based signal (Em525), we agree that the *previous Fig 2b-e* showed an exaggerated Em460 decrease and an underrated Em525 increase. We suppose that some experimental conditions influenced the estimation of these signals. When acquiring microscopic images, we often experienced variations of signal intensities under various experimental conditions, including objective lenses, MAM-Calflux expression levels, and substrate (furimazine) concentrations. These differences in signal intensities might cause different signal-to-noise ratios. However, images were processed by subtracting background signals before quantitation, which

might have affected the ratio of changes by IP₃ or histamine treatment. In addition, when cell density and transfection rate were particularly high, we experienced a fast decrease of Em460 and Em525 signals, possibly due to the increased substrate exhaustion. The rapid decline of signals may have exaggerated the drop of Em460 signals and blunted the increase of Em525 signals. Thus, we put extra care into setting the experimental conditions to control sample-by-sample or set-by-set variations within a narrow range.

→ In response to the comment, we replaced **Fig 2a-g** with newly performed experimental data, demonstrating an apparent increase of Em525 and a decrease of Em460. In this experimental setting, we carefully controlled cell density, transfection rate, imaging buffers volume, and ROI selection for this time-lapse imaging. Accordingly, we revised the Methods section corresponding to these experimental details (*page 16, lines 375-380 and 389-391, page 18, lines 436-444, and page 19, lines 468-471*).

Fig 2a-g

3. Do you have a better example image for Figure 2a? In this example, it looks like the Em525 decreases immediately after IP₃ treatment, which is not representative of panel e.

→ In response to the reviewer's suggestion, we have replaced the representative image in **Fig 2a**.

References

- 1 Szabadkai, G. *et al.* Chaperone-mediated coupling of endoplasmic reticulum and mitochondrial Ca²⁺ channels. *J Cell Biol* **175**, 901-911 (2006). <https://doi.org/10.1083/jcb.200608073>
- 2 Tiwary, S., Nandwani, A., Khan, R. & Datta, M. GRP75 mediates endoplasmic reticulum-mitochondria coupling during palmitate-induced pancreatic beta-cell apoptosis. *J Biol Chem* **297**, 101368 (2021). <https://doi.org/10.1016/j.jbc.2021.101368>
- 3 Lee, S., Wang, W., Hwang, J., Namgung, U. & Min, K. T. Increased ER-mitochondria tethering promotes axon regeneration. *Proc Natl Acad Sci U S A* **116**, 16074-16079 (2019). <https://doi.org/10.1073/pnas.1818830116>
- 4 Honrath, B. *et al.* Glucose-regulated protein 75 determines ER-mitochondrial coupling and sensitivity to oxidative stress in neuronal cells. *Cell Death Discov* **3**, 17076 (2017). <https://doi.org/10.1038/cddiscovery.2017.76>
- 5 de Brito, O. M. & Scorrano, L. Mitofusin 2 tethers endoplasmic reticulum to mitochondria. *Nature* **456**, 605-610 (2008). <https://doi.org/10.1038/nature07534>
- 6 Schneeberger, M. *et al.* Mitofusin 2 in POMC neurons connects ER stress with leptin resistance and energy imbalance. *Cell* **155**, 172-187 (2013). <https://doi.org/10.1016/j.cell.2013.09.003>
- 7 Naon, D. *et al.* Critical reappraisal confirms that Mitofusin 2 is an endoplasmic reticulum-mitochondria tether. *Proc Natl Acad Sci U S A* **113**, 11249-11254 (2016). <https://doi.org/10.1073/pnas.1606786113>
- 8 Han, S. *et al.* The role of Mfn2 in the structure and function of endoplasmic reticulum-mitochondrial tethering in vivo. *J Cell Sci* **134** (2021). <https://doi.org/10.1242/jcs.253443>
- 9 Casellas-Diaz, S. *et al.* Mfn2 localization in the ER is necessary for its bioenergetic function and neuritic development. *EMBO Rep* **22**, e51954 (2021). <https://doi.org/10.15252/embr.202051954>
- 10 Hutchins, B. I. & Kalil, K. Differential outgrowth of axons and their branches is regulated by localized calcium transients. *J Neurosci* **28**, 143-153 (2008). <https://doi.org/10.1523/JNEUROSCI.4548-07.2008>
- 11 Spillane, M., Ketschek, A., Merianda, T. T., Twiss, J. L. & Gallo, G. Mitochondria coordinate sites of axon branching through localized intra-axonal protein synthesis. *Cell Rep* **5**, 1564-1575 (2013). <https://doi.org/10.1016/j.celrep.2013.11.022>
- 12 Wu, Y. *et al.* Contacts between the endoplasmic reticulum and other membranes in neurons. *Proc Natl Acad Sci U S A* **114**, E4859-E4867 (2017). <https://doi.org/10.1073/pnas.1701078114>
- 13 Bodakuntla, S., Nedostralova, H., Basnet, N. & Mizuno, N. Cytoskeleton and Membrane Organization at Axon Branches. *Front Cell Dev Biol* **9**, 707486 (2021). <https://doi.org/10.3389/fcell.2021.707486>
- 14 Yang, J. *et al.* Coupling optogenetic stimulation with NanoLuc-based luminescence (BRET) Ca(++) sensing. *Nat Commun* **7**, 13268 (2016). <https://doi.org/10.1038/ncomms13268>
- 15 Hirabayashi, Y. *et al.* ER-mitochondria tethering by PDZD8 regulates Ca(2+) dynamics in mammalian neurons. *Science* **358**, 623-630 (2017). <https://doi.org/10.1126/science.aan6009>
- 16 Li, J. *et al.* GRP75-facilitated Mitochondria-associated ER Membrane (MAM) Integrity controls Cisplatin-resistance in Ovarian Cancer Patients. *Int J Biol Sci* **18**, 2914-2931 (2022). <https://doi.org/10.7150/ijbs.71571>
- 17 Zampese, E. *et al.* Presenilin 2 modulates endoplasmic reticulum (ER)-mitochondria interactions and Ca²⁺ cross-talk. *Proc Natl Acad Sci U S A* **108**, 2777-2782 (2011). <https://doi.org/10.1073/pnas.1100735108>

- 18 Stoica, R. *et al.* ER-mitochondria associations are regulated by the VAPB-PTPIP51 interaction and are disrupted by ALS/FTD-associated TDP-43. *Nat Commun* **5**, 3996 (2014). <https://doi.org:10.1038/ncomms4996>
- 19 Gomez-Suaga, P. *et al.* The ER-Mitochondria Tethering Complex VAPB-PTPIP51 Regulates Autophagy. *Curr Biol* **27**, 371-385 (2017). <https://doi.org:10.1016/j.cub.2016.12.038>
- 20 De Vos, K. J. *et al.* VAPB interacts with the mitochondrial protein PTPIP51 to regulate calcium homeostasis. *Hum Mol Genet* **21**, 1299-1311 (2012). <https://doi.org:10.1093/hmg/ddr559>
- 21 Lalier, L. *et al.* TOM20-mediated transfer of Bcl2 from ER to MAM and mitochondria upon induction of apoptosis. *Cell Death Dis* **12**, 182 (2021). <https://doi.org:10.1038/s41419-021-03471-8>
- 22 Simmen, T. *et al.* PACS-2 controls endoplasmic reticulum-mitochondria communication and Bid-mediated apoptosis. *EMBO J* **24**, 717-729 (2005). <https://doi.org:10.1038/sj.emboj.7600559>
- 23 Giorgi, C. *et al.* p53 at the endoplasmic reticulum regulates apoptosis in a Ca²⁺-dependent manner. *Proc Natl Acad Sci U S A* **112**, 1779-1784 (2015). <https://doi.org:10.1073/pnas.1410723112>
- 24 Bonneau, B. *et al.* IRBIT controls apoptosis by interacting with the Bcl-2 homolog, Bcl2l10, and by promoting ER-mitochondria contact. *Elife* **5** (2016). <https://doi.org:10.7554/eLife.19896>
- 25 Yang, Z., Zhao, X., Xu, J., Shang, W. & Tong, C. A novel fluorescent reporter detects plastic remodeling of mitochondria-ER contact sites. *J Cell Sci* **131** (2018). <https://doi.org:10.1242/jcs.208686>
- 26 Hertlein, V. *et al.* MERLIN: a novel BRET-based proximity biosensor for studying mitochondria-ER contact sites. *Life Sci Alliance* **3** (2020). <https://doi.org:10.26508/lsa.201900600>
- 27 Chen, L., Xie, Z., Turkson, S. & Zhuang, X. A53T human alpha-synuclein overexpression in transgenic mice induces pervasive mitochondria macroautophagy defects preceding dopamine neuron degeneration. *J Neurosci* **35**, 890-905 (2015). <https://doi.org:10.1523/JNEUROSCI.0089-14.2015>
- 28 Choubey, V. *et al.* Mutant A53T alpha-synuclein induces neuronal death by increasing mitochondrial autophagy. *J Biol Chem* **286**, 10814-10824 (2011). <https://doi.org:10.1074/jbc.M110.132514>
- 29 Vicario, M., Cieri, D., Brini, M. & Cali, T. The Close Encounter Between Alpha-Synuclein and Mitochondria. *Front Neurosci* **12**, 388 (2018). <https://doi.org:10.3389/fnins.2018.00388>
- 30 Xu, X. *et al.* Imaging protein interactions with bioluminescence resonance energy transfer (BRET) in plant and mammalian cells and tissues. *Proc Natl Acad Sci U S A* **104**, 10264-10269 (2007). <https://doi.org:10.1073/pnas.0701987104>
- 31 Cumberbatch, D. *et al.* A BRET Ca²⁺ sensor enables high-throughput screening in the presence of background fluorescence. *Sci Signal* **15**, eabq7618 (2022). <https://doi.org:10.1126/scisignal.abq7618>
- 32 Ohashi, K., Kiuchi, T., Shoji, K., Sampei, K. & Mizuno, K. Visualization of cofilin-actin and Ras-Raf interactions by bimolecular fluorescence complementation assays using a new pair of split Venus fragments. *Biotechniques* **52**, 45-50 (2012). <https://doi.org:10.2144/000113777>
- 33 Kodama, Y. & Hu, C. D. An improved bimolecular fluorescence complementation assay with a high signal-to-noise ratio. *Biotechniques* **49**, 793-805 (2010). <https://doi.org:10.2144/000113519>

- 34 Nakagawa, C., Inahata, K., Nishimura, S. & Sugimoto, K. Improvement of a Venus-based bimolecular fluorescence complementation assay to visualize bFos-bJun interaction in living cells. *Biosci Biotechnol Biochem* **75**, 1399-1401 (2011). <https://doi.org:10.1271/bbb.110189>
- 35 Alford, S. C., Ding, Y., Simmen, T. & Campbell, R. E. Dimerization-dependent green and yellow fluorescent proteins. *ACS Synth Biol* **1**, 569-575 (2012). <https://doi.org:10.1021/sb300050j>
- 36 Orioka, M. *et al.* A Series of Furimazine Derivatives for Sustained Live-Cell Bioluminescence Imaging and Application to the Monitoring of Myogenesis at the Single-Cell Level. *Bioconjug Chem* **33**, 496-504 (2022). <https://doi.org:10.1021/acs.bioconjchem.2c00035>

REVIEWERS' COMMENTS

Reviewer #1 (Remarks to the Author):

The authors have made a solid effort to answer my queries and I can support the publication of the manuscript in its present form.

Reviewer #2 (Remarks to the Author):

Cho and co-workers extensively addressed all the criticisms this referee raised in the manuscript revision process.

The revised manuscript provides solid evidence that the recombinant probe developed can report the concentration of Ca^{2+} at MAMs.

Before recommending the manuscript to be accepted for publication, I need to point out a minor comment that I believe can significantly improve the manuscript.

In more than one figure, the authors report that the co-administration of Ip_3 and Ionomycin can elicit the release of Ca^{2+} from the IP3R. Experimental artifacts must affect this data and could lead future readers to the wrong conclusion.

This consideration is based on two facts:

- Ip_3 is highly hydrophobic and cannot diffuse across the plasma membrane when administered to integer cells; it cannot reach the IP3R to induce its opening.
- Ionomycin is a Ca^{2+} ionophore and does not make the Ip_3 permeable to the plasma membrane, regardless of the high concentration used. On the other hand, Ionomycin equilibrates $[\text{Ca}^{2+}]$ across all membranes. Since the experiment runs in Ca^{2+} free extracellular buffer, Ionomycin administration would cause the transient release of Ca^{2+} across the ER membrane to all intracellular compartments, ultimately leading to its diffusion in the extracellular bath.

Given these conditions, the elevation of $[\text{Ca}^{2+}]$ measured by the MAM-Calflux, in cells co-stimulated by Ip_3 /Ionomycin must be independent of IP3R and MAMs.

I strongly recommend removing these data from the manuscript as these can lead to misinterpretation also because the validation of the probe is backed up by other data.

Reviewer #3 (Remarks to the Author):

In this revised manuscript, Cho et al. have performed new crucial characterization of their MAM-calflux sensor (Figure 1g-h), repeated several of their experiments using new considerations for expression levels (Figure 2b-e), re-analyzed existing data using a more stringent masking methodology, and also included significant new data in Figure 4, highlighting the capabilities of MAM-Calflux in measuring subcellular dynamics at branch points of developing neurites. My specific prior major concerns were met with new experiments. They successfully demonstrated that the expression alone of MAM-Calflux does not drive large-scale structural changes at shorter expression levels (using ER-mito colocalization analysis). They appropriately acknowledge that upon longer-term expression levels (48 hours), this can drive mito-ER contacts in a non-physiological manner (as is apparent with most split-proteins). They acknowledge this explicitly in the methods, as appropriate. This new data also addressed my concern about expression level variabilities. Finally the new data show in Figure 4 is much stronger and rigorous at the population level compared to the prior data shown in Figure 4, providing another example of the tool's utility. In summary, I believe the authors have demonstrated significant utility of this new technology, while highlighting the important caveats, and I recommend publication of this manuscript.

Response to Reviewer #2 (Remarks to the Author):

Cho and co-workers extensively addressed all the criticisms this referee raised in the manuscript revision process.

The revised manuscript provides solid evidence that the recombinant probe developed can report the concentration of Ca²⁺ at MAMs.

Before recommending the manuscript to be accepted for publication, I need to point out a minor comment that I believe can significantly improve the manuscript.

In more than one figure, the authors report that the co-administration of Ip₃ and Ionomycin can elicit the release of Ca²⁺ from the IP₃R. Experimental artifacts must affect this data and could lead future readers to the wrong conclusion.

This consideration is based on two facts:

- Ip₃ is highly hydrophobic and cannot diffuse across the plasma membrane when administered to integer cells; it cannot reach the IP₃R to induce its opening.

- Ionomycin is a Ca²⁺ ionophore and does not make the Ip₃ permeable to the plasma membrane, regardless of the high concentration used. On the other hand, Ionomycin equilibrates [Ca²⁺] across all membranes. Since the experiment runs in Ca²⁺ free extracellular buffer, Ionomycin administration would cause the transient release of Ca²⁺ across the ER membrane to all intracellular compartments, ultimately leading to its diffusion in the extracellular bath.

Given these conditions, the elevation of [Ca²⁺] measured by the MAM-Caflux, in cells co-stimulated by Ip₃/Ionomycin must be independent of IP₃R and MAMs.

I strongly recommend removing these data from the manuscript as these can lead to misinterpretation also because the validation of the probe is backed up by other data.

→ In response to the reviewer's recommendation, we removed all data using IP₃/ionomycin co-administration. Accordingly, we revised the corresponding Results and Methods sections (*page 6, lines 137-145, and page 10, line 237*).

→ In **Fig 2**, we removed IP₃/ionomycin treatment results (*previous Fig 2a-g*). Thus, histamine and CaCl₂/ionomycin treatment results (*previous Fig 2h-l*) became **Fig 2a-e**.

→ In **Fig 6**, we removed IP₃/ionomycin-induced Ca²⁺ imaging results from SNCA*A53T neurons (*previous Fig 6e-j*). Instead, we moved the *previous Supplementary fig 11*, in which we introduced the VAPB-PTPIP51 complex to SNCA*A53T tg neurons to restore MAM structures, to **Fig 6e-I**. We believe this part gets along well with the other panels in Figure 6 by further describing the nature of MAM Ca²⁺ dynamics SNCA*A53T tg neurons.

Fig 2

Fig 6